# AVAILABILITY ATTACKS NEED TO CREATE SHORTCUTS FOR CONTRASTIVE LEARNING

## ABSTRACT

Availability attacks can prevent the unauthorized use of private data and commercial datasets by generating imperceptible noise and making unlearnable examples before release. Ideally, the obtained unlearnability prevents algorithms from training usable models. When supervised learning algorithms have failed, a malicious data collector possibly resorts to contrastive learning algorithms to bypass the protection. Attacks need both supervised unlearnability and contrastive unlearnability. Through evaluation, we have found that most of the existing availability attacks are unable to achieve contrastive unlearnability, which poses risks to data protection. Furthermore, we find that employing stronger data augmentations in supervised poisoning generation can create contrastive shortcuts and mitigate this risk. Based on this insight, we propose AUE and AAP attacks which prominently boost the worst-case unlearnability across multiple supervised and contrastive algorithms.

## 1 INTRODUCTION

Availability attacks (Biggio & Roli, 2018) add imperceptible poisons to training data such that a subsequently trained model becomes unavailable. The motivations behind these attack methods involve protecting private data and commercial datasets from unauthorized use. For example, a malicious data collector may gather selfies from social media applications into a training set. Based on a model trained on this data, individual identities can be inferred from future street photos or surveillance images. In this type of scenario, availability attacks provide tools to process user images before release such that processed images remain legible but hinder subsequent training.

In recent years, various availability attacks have been proposed (Feng et al., 2019; Huang et al., 2020; Fowl et al., 2021). These approaches successfully suppress the performance of a supervised model below a usable level. Meanwhile, contrastive learning algorithms have achieved comparable performance to supervised algorithms (Chen et al., 2020a;b; Grill et al., 2020; Chen & He, 2021). Thus, an unauthorized data collector can use contrastive learning to train a model. Recently, supervised poisoning frameworks were extended to poison contrastive learning (He et al., 2022; Ren et al., 2022).

In Figure 1, we conduct an assessment of both supervised and contrastive unlearnability of existing availability attacks. The abbreviation for attacks can be found in Section 4. Most attacks designed for poisoning supervised learning can not handle contrastive learning. These findings shed light on a potential issue of using availability attacks to protect data: a malicious data collector can traverse both supervised and contrastive algorithms to effectively leverage collected data. While successful attacks for supervised learning leverage linear separable noises as shortcuts (Yu et al., 2022), we find that contrastive unlearnability requires huge alignment and uniformity gaps between poisoned data and clean data as shortcuts. *A fully functional availability attack should create shortcuts for both supervised learning and contrastive learning.* To clarify this issue, we propose a threat model that considers the worst-case unlearnability across supervised and contrastive algorithms.

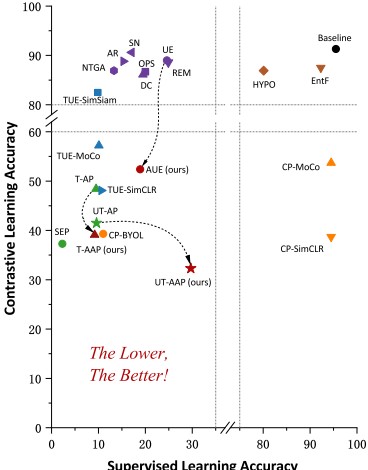

Figure 1: Supervised learning and SimCLR on CIFAR-10 under availability attacks.

Error-minimizing noises (Huang et al., 2020) and adversarial poisoning (Fowl et al., 2021) are two representative supervised availability attacks that craft noises using reference models. While supervised learning uses mild data augmentations, contrastive learning augmentations are much stronger. We find that supervised training of reference models with enhanced data augmentations can mimic contrastive training. Furthermore, by crafting noises on these contrastive-like reference models through enhanced data augmentations in addition, the resulting attack can adapt to contrastive augmentations and possess shortcuts for both contrastive learning and supervised learning. Based on this, we propose an augmented unlearnable example attack (AUE) and an augmented adversarial poisoning attack (AAP) which achieve better worst-case unlearnability across multiple algorithms compared to existing methods. We summarize our contributions:

- We point out that contrastive learning algorithms can undermine data protection using availability attacks and introduce worst-case unlearnability across supervised and contrastive algorithms as an evaluation metric for availability attacks.

- We analyze the shortcuts for contrastive learning and reveal that enhanced data augmentations can boost the contrastive unlearnability of basic supervised approaches.

- Our proposed AUE and AAP attacks improve worst-case unlearnability across five algorithms on CIFAR-10/CIFAR-100 compared to existing baselines and are effective on Tiny-ImageNet, Mini-ImageNet, and ImageNet-100.

## 2 RELATED WORKS

Availability attacks for supervised learning include error-minimizing noises(Huang et al., 2020), adversarial example poisoning (Fowl et al., 2021; Chen et al., 2023), neural tangent generalization attack (Yuan & Wu, 2021), generative poisoning attack (Feng et al., 2019), autoregression perturbation (Sandoval-Segura et al., 2022), one-pixel perturbation perturbation Wu et al. (2022), convolution-based attack Sadasivan et al. (2023) and synthetic perturbation (Yu et al., 2022). Robust error-minimizing errors (Fu et al., 2022), entangled features strategy (Wen et al., 2023), and hypocritical perturbation (Tao et al., 2022) are proposed to deceive adversarial training. Contrastive poisoning (He et al., 2022) and transferable unlearnable examples (Ren et al., 2022) aim at poisoning contrastive learning. Unlearnable clusters (Zhang et al., 2023) proposed to generate label-agnostic noises with cluster-wise perturbations. On the defense side, adversarial training can largely mitigate the unlearnablity (Tao et al., 2021). Liu et al. (2023); Qin et al. (2023) leverages crafted data augmentations to defend against availability attacks. Sandoval-Segura et al. (2023) suggests that the orthogonal projection technique can effectively defend against class-wise attacks. Diffusion models are used to purify unlearnable perturbations (Jiang et al., 2023; Dolatabadi et al., 2023).

## 3 THREAT MODEL AND BACKGROUNDS

### 3.1 FORMAL THREAT MODEL

In our threat model, we assume that an unauthorized data collector assembles labeled data into a dataset. The access to label information is reasonable since the collector can crawl individual images from certain accounts or steal (and annotate) a commercial dataset. A data publisher is supposed to process data before release using an availability attack such that processed data is resilient to subsequent supervised algorithms as well as contrastive algorithms adopted by the data collector.

In general, a finite dataset $\mathcal{D}_c$ that needs to be processed is *i.i.d* sampled from a data distribution $\mathcal{D}$. For a data-label pair $(\boldsymbol{x}, y) \in \mathcal{D}_c$, an availability attack $\delta$ maps it to a noise $\delta(\boldsymbol{x}, y)$ within a $L_p$-norm ball $\mathcal{B}_p(\epsilon)$. In this paper, we set $p = \infty$ and radius $\epsilon = 8/255$. It results in a protected dataset $\{\boldsymbol{x} + \delta(\boldsymbol{x}, y) | (\boldsymbol{x}, y) \in \mathcal{D}_c\}$ to which a data collector has only access. For potential algorithms, we refer $f$ to a supervised model and $g$ to a contrastive feature extractor beyond which is a linear probing head $h$. The goal of the data publisher is to find a poisoning map $\delta$ that significantly degrades the generalization performance of both $f_\delta$ and $h_\delta \circ g_\delta$ which are well trained on poisoned data. When we consider the *worst-case unlearnability across supervised and contrastive algorithms*, the threat

model has the following mathematical form:

$$\min_{\delta} \max(\mathbb{E}_{\mathcal{D}}\left[\mathbf{1}(f_{\delta}(\boldsymbol{x}) = y)\right], \mathbb{E}_{\mathcal{D}}\left[\mathbf{1}(h_{\delta} \circ g_{\delta}(\boldsymbol{x}) = y)\right]) \quad (1)$$

$$\text{s.t.} \quad f_{\delta} \in \arg\min_{f} \mathbb{E}_{\mathcal{D}_c}\left[\mathcal{L}_{\text{SL}}(\boldsymbol{x} + \delta(\boldsymbol{x}, y), y; f)\right], \qquad \textit{(supervised learning)}$$

$$g_{\delta} \in \arg\min_{g} \mathbb{E}_{\mathcal{D}_c}\left[\mathcal{L}_{\text{CL}}(\boldsymbol{x} + \delta(\boldsymbol{x}, y); g)\right], \qquad \textit{(contrastive learning)}$$

$$h_{\delta} \in \arg\min_{h} \mathbb{E}_{\mathcal{D}_c}\left[\mathcal{L}_{\text{SL}}(\boldsymbol{x} + \delta(\boldsymbol{x}, y), y; h \circ g_{\delta})\right]. \qquad \textit{(linear probing)}$$

Here we denote $\mathcal{L}_{\text{CL}}(\cdot; \cdot)$ as contrastive loss for simplicity, but in practice, it usually involves one positive sample and several negative samples. When a poisoning map $\delta(\boldsymbol{x}, y)$ only depends on label $y$, the resulting attack is called a class-wise attack; otherwise, it is a sample-wise attack. In this paper, we mainly focus on sample-wise attacks if not otherwise stated. In the contrastive part of our threat model, there is a difference from the setting adopted by He et al. (2022); Ren et al. (2022) in which the linear probing stage relies on the unprocessed clean data as downstream tasks. We have more discussion in Appendix B.7.

## 3.2 BASIC APPROACHES

The essence of availability attacks is to prevent a trained model from well generalizing to the clean data distribution. It has been revealed that linearly separable patterns in crafted noises work as shortcuts for the training process of supervised learning (Yu et al., 2022).

**Error-minimizing noises.** Unlearnable example attacks (UE, Huang et al. (2020)) generate noises by alternately optimizing a bi-level problem:

$$\min_{\delta} \min_{f} \mathbb{E}_{\mathcal{D}_c}\left[\mathcal{L}_{\text{SL}}(\boldsymbol{x} + \delta(\boldsymbol{x}, y), y; f)\right]. \quad (2)$$

The error-minimization framework has been extended to contrastive settings (CP, He et al. (2022)):

$$\min_{\delta} \min_{g} \mathbb{E}_{\mathcal{D}_c}\left[\mathcal{L}_{\text{CL}}(\boldsymbol{x} + \delta(\boldsymbol{x}, y); g)\right]. \quad (3)$$

Then a regularization term called class-wise separability discriminant (CSD) was introduced to equip noises with linear-separable shortcuts for supervised learning (TUE, Ren et al. (2022)).

**Adversarial poisoning.** Generated by PGD-Attack (Madry et al., 2018) on a pre-trained reference model, adversarial examples can make strong poisons (AP, Fowl et al. (2021)):

$$\min_{\delta} \mathbb{E}_{\mathcal{D}_c}\left[\mathcal{L}_{\text{SL}}(\boldsymbol{x} + \delta(\boldsymbol{x}, y), y + K; f^*)\right] \qquad \textit{(Targeted)}$$

$$or \quad \max_{\delta} \mathbb{E}_{\mathcal{D}_c}\left[\mathcal{L}_{\text{SL}}(\boldsymbol{x} + \delta(\boldsymbol{x}, y), y; f^*)\right] \qquad \textit{(Untargeted)} \quad (4)$$

$$\text{s.t.} \quad f^* \in \arg\min_{f} \mathbb{E}_{\mathcal{D}_c}\left[\mathcal{L}_{\text{SL}}(\boldsymbol{x}, y; f)\right].$$

Adversarial poisoning assigns clean data with a crafted noise containing non-robust but useful features of another label that confound learning algorithms. Recently, Chen et al. (2023) proposed self-ensemble protection (SEP-FA-VR) that generated adversarial poisons using several checkpoints to improve supervised unlearnability.

## 3.3 CONTRASTIVE LEARNING

Contrastive learning (CL) is self-supervised and does not require label information until linear probing. In general, it first augments an input into two views using augmentations sampled from a strong augmentation distribution $\mu$. Then extracted features are trained to be aligned between positive pairs (views of the same input) but distinct between negative pairs (views of different inputs).

Wang & Isola (2020) introduced two key properties for contrastive learning, *alignment* and *uniformity*. The former measures the similarity of features from positive pairs and the latter reflects the uniformity of feature distribution on the hypersphere. Let $g$ be a normalized feature extractor and $\mu$ be a data augmentation distribution. The *alignment loss* and *uniformity loss* are defined as the following:

$$\mathcal{A}(\mathcal{D}_c) = \mathop{\mathbb{E}}_{\substack{\boldsymbol{x} \sim \mathcal{D}_c \\ \pi, \tau \sim \mu}}\left[||g(\pi(\boldsymbol{x})) - g(\tau(\boldsymbol{x}))||_2^2\right], \quad \mathcal{U}(\mathcal{D}_c) = \log \mathop{\mathbb{E}}_{\substack{\boldsymbol{x}, \boldsymbol{z} \sim \mathcal{D}_c \\ \pi, \tau \sim \mu}}\left[e^{-2||g(\pi(\boldsymbol{x})) - g(\tau(\boldsymbol{z}))||_2^2}\right].$$

## 4 SHORTCUTS FOR CONTRASTIVE LEARNING

Let $\mathcal{D}_c'$ be a poisoned dataset with respect to a clean dataset $\mathcal{D}_c$. The *alignment gap* and *uniformity gap* between clean and poisoned datasets are defined as follows:

$$\mathcal{AG}(\mathcal{D}_c, \mathcal{D}_c') = \mathcal{A}(\mathcal{D}_c) - \mathcal{A}(\mathcal{D}_c'), \qquad \mathcal{UG}(\mathcal{D}_c, \mathcal{D}_c') = \mathcal{U}(\mathcal{D}_c) - \mathcal{U}(\mathcal{D}_c').$$

We train ResNet-18 models on the CIFAR-10 training sets poisoned by various attacks using SimCLR (Chen et al., 2020a) without linear probing or fine-tuning. These poisoning attacks include UE (Huang et al., 2020), DC (Feng et al., 2019), NTGA (Yuan & Wu, 2021), SN (Yu et al., 2022), HYPO (Tao et al., 2022), EntF (Wen et al., 2023), AR (Sandoval-Segura et al., 2022), OPS (Wu et al., 2022), REM (Fu et al., 2022), AP (Fowl et al., 2021), SEP-FA-VR (Chen et al., 2023), CP (He et al., 2022), and TUE (Ren et al., 2022).

In Table 1, we evaluate alignment and uniformity gaps between clean and poisoned datasets, as well as the SimCLR accuracy and SL accuracy. On one hand, AP-based attacks achieve both SL and SimCLR unlearnability while other non-contrastive poisoning attacks fail to deceive the contrastive learning algorithm. The alignment and uniformity gaps of AP-based attacks are prominently larger than those of others. On the other hand, contrastive error-minimizing attacks including CP and TUE are effective for contrastive learning and possess huge alignment and uniformity gaps.

The Pearson correlation coefficient (PCC) between the alignment gap and the SimCLR accuracy is -0.78, and the PCC between the uniformity gap and the SimCLR accuracy is -0.87. It is revealed that contrastive unlearnability highly relates to huge alignment and uniformity gaps which indicate a significant

Table 1: Shortcuts on SimCLR models trained on poisoned CIFAR-10. **Bold** fonts emphasize prominent contrastive unlearnability values.

| Attack | $\mathcal{AG}$ | $\mathcal{UG}$ | SimCLR | SL |
|---|---|---|---|---|
| DC | 0.12 | 0.07 | 86.1 | 19.5 |
| UE | 0.05 | 0.03 | 89.0 | 24.6 |
| AR | 0.07 | 0.09 | 88.8 | 15.3 |
| NTGA | 0.12 | 0.12 | 86.9 | 13.3 |
| SN | 0.08 | 0.00 | 90.6 | 17.1 |
| OPS | 0.04 | 0.01 | 86.7 | 20.0 |
| REM | 0.12 | 0.04 | 88.6 | 24.9 |
| EntF | 0.01 | -0.04 | 87.5 | 92.3 |
| HYPO | 0.11 | 0.13 | 86.9 | 80.1 |
| T-AP | **0.18** | **0.44** | **48.4** | 9.5 |
| UT-AP | **0.17** | **0.77** | **41.5** | 9.6 |
| SEP-FA-VR | **0.24** | **0.25** | **37.3** | 2.3 |
| CP-SimCLR | **0.55** | **0.87** | **38.7** | 94.5 |
| TUE-SimCLR | **0.30** | **0.76** | **48.1** | 10.6 |

difference between clean feature distribution and poisoned feature distribution. The gaps work as shortcuts for CL. After the poisoned unsupervised training, the feature extractor is fixed and the linear probing stage trains a linear layer to classify poisoned features. If the gaps are huge, even though the extracted features of poisoned data are highly linear separable, the learned separability can hardly be generalized to clean features due to the huge discrepancy between clean features and poisoned features. Consequently, even if the accuracy of poisoned data is high, the accuracy of clean data is low and the poisoning attack is successful. On the contrary, small gaps likely imply clean features are similar to poisoned features. Therefore, once the classifier can perform correct classification on poisoned data, it can generalize to clean data and thus the availability attack fails.

Since contrastive error-minimizing noises directly minimize the contrastive loss on poisoned data which relates to alignment and uniformity, they naturally create prominent gaps as shortcuts for CL (He et al., 2022). In the next section, we will demonstrate that supervised error minimization with enhanced data augmentations can partially replace the role of contrastive error minimization to enlarge the alignment and uniformity gaps and bring contrastive unlearnability. Moreover, we find stronger augmentations help adversarial poisoning generate poisons to deceive a contrastive-like reference model and thus improve the contrastive unlearnability as well.

## 5 ENHANCED DATA AUGMENTATIONS BOOST BASIC APPROACHES

### 5.1 MIMIC CONTRASTIVE LEARNING WITH SUPERVISED MODELS

Contrastive learning employs strong data augmentations including resized crop, color jitter, horizontal flip, and grayscale. Supervised learning adopts mild data augmentations such as horizontal flip and crop to improve generalization, but excessive data augmentations can harm performance. In Appendix A.2, Pesudo-Code 1 shows the detailed implementations for these two different settings.

Specifically, a strength hyperparameter $s \in [0, 1]$ is introduced to control the intensity of contrastive augmentations. A large $s$ stands for strong augmentations, and it is set $s = 1$ by default for contrastive learning. Non-contrastive availability attacks whose noise generation involves optimizing supervised losses, usually inherit mild augmentations. To make noise possess unlearnable features that can withstand contrastive augmentations, a natural approach is to incorporate strong data augmentations during their generation. We find that with enhanced data augmentations, noises generated by supervised methods do not only adapt to contrastive augmentations but also learn to deal with contrastive loss.

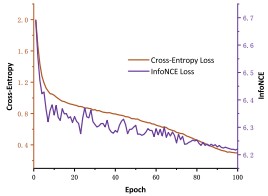

Figure 2: InfoNCE loss decreases with CE loss.

On CIFAR-10, we train a ResNet-18 for 100 epochs using data augmented by contrastive augmentations, i.e. augmentation strength $s = 1$. For each checkpoint, the supervised cross-entropy loss and the contrastive InfoNCE loss (Oord et al., 2018) are computed on the training set. In Figure 2, when the optimization object CE loss goes down, the InfoNCE loss decreases as well. It illustrates that the supervised training with contrastive augmentations implicitly optimizes the contrastive loss. Since supervised models can mimic contrastive learning to some extent, incorporating stronger data augmentation can potentially enable poisoning attacks based on supervised training to acquire the ability to deceive contrastive learning.

To have a closer look at the relationship between supervised loss and contrastive loss, we study a toy model $f = h \circ g : \mathbb{R}^d \to \mathbb{R}^n$ with a normalized feature extractor $g : \mathbb{R}^d \to \mathbb{R}^n$ such that $||g(\boldsymbol{x})|| \equiv 1$ and a full rank linear classifier $h : \mathbb{R}^n \to \mathbb{R}^n$ in the sense that $h(\boldsymbol{z}) = W\boldsymbol{z} + \boldsymbol{b}$ with a full rank square matrix $W \in \mathbb{R}^{n \times n}$. By singular values decomposition (SVD), $W = U\Sigma V$ with orthogonal matrices $U, V \in \mathbb{R}^{n \times n}$ and $\Sigma = \text{diag}(\sigma_1, \cdots, \sigma_n), \sigma_1 \geq \cdots \geq \sigma_n > 0$. Let $\mathcal{D}$ be a balanced data distribution, i.e. each class would be sampled with the same probability, $\mathcal{D}_x$ be the margin distribution, and $\mu$ be an augmentation distribution. Assume the supervised loss $\mathcal{L}_{\text{SL}}$ is the mean squared error and the contrastive loss $\mathcal{L}_{\text{CL}}$ contains only one negative example:

$$\mathcal{E}_{\text{SL}} = \mathop{\mathbb{E}}_{\substack{(\boldsymbol{x},y)\sim\mathcal{D} \\ \pi\sim\mu}} \big[ \mathcal{L}_{\text{SL}}(\boldsymbol{x}, y, \pi) \big] = \mathop{\mathbb{E}}_{\substack{(\boldsymbol{x},y)\sim\mathcal{D} \\ \pi\sim\mu}} \big[ \frac{1}{n} ||h \circ g(\pi(\boldsymbol{x})) - \boldsymbol{e}_y||^2 \big],$$

$$\mathcal{E}_{\text{CL}} = \mathop{\mathbb{E}}_{\substack{\boldsymbol{x},\boldsymbol{x}^-\sim\mathcal{D}_x \\ \pi,\tau,\rho\sim\mu}} \big[ \mathcal{L}_{\text{CL}}(\boldsymbol{x}, \boldsymbol{x}^-, \pi, \tau, \rho) \big] = \mathop{\mathbb{E}}_{\substack{\boldsymbol{x},\boldsymbol{x}^-\sim\mathcal{D}_x \\ \pi,\tau,\rho\sim\mu}} \big[ -\log \frac{e^{g(\pi(\boldsymbol{x}))^\top g(\tau(\boldsymbol{x}))}}{e^{g(\pi(\boldsymbol{x}))^\top g(\tau(\boldsymbol{x}))} + e^{g(\pi(\boldsymbol{x}))^\top g(\rho(\boldsymbol{x}^-))}} \big].$$

The following theorem says the upper bound of contrastive loss $\mathcal{L}_{\text{CL}}$ decreases as the supervised loss $\mathcal{L}_{\text{SL}}$ decreases in a range of values if their data augmentations obey the same distribution. .

**Theorem 5.1.** *With probability at least $1 - 4\sqrt{\mathcal{E}_{\text{SL}}}$, it holds that*

$$\mathcal{L}_{\text{CL}}(\boldsymbol{x}, \boldsymbol{x}^-, \pi, \tau, \rho) < \frac{n-1}{n} \log(1 + \frac{\sigma_1^2\sigma_n - \sigma_n(1 - \sqrt{2n\sqrt{\mathcal{E}_{\text{SL}}}})^2}{\sigma_1^2\sigma_n - 2n\sigma_1^2\sqrt{\mathcal{E}_{\text{SL}}}}) + \frac{1}{n}\log(1 + \frac{\sigma_n}{\sigma_n - 2n\sqrt{\mathcal{E}_{\text{SL}}}}).$$

*Remark* 5.2. 1) Assumptions of a square matrix and positive singular values are necessary. Otherwise, the dimensional reduction of feature space impairs the relation between supervised and contrastive losses. 2) Since supervised losses contain limited information about negative pairs, this inequality is naturally loose. However, in the case that supervised learning fits very well, it at least implies that positive features $g(\tau(\boldsymbol{x}))$ are closer to $g(\pi(\boldsymbol{x}))$ than negative features $g(\rho(\boldsymbol{x}^-))$.

### 5.2 AUGMENTED UNLEARNABLE EXAMPLES (AUE)

Unlearnable examples are generated by supervised error minimization in which a reference model and noises alternately update in Equ. 2. Previous results imply that the contrastive loss is upper bound by the supervised loss when the supervised learning adopts the same contrastive augmentations. Specifically, when noises are added in a differentiable way, i.e. $\pi(\boldsymbol{x} + \delta(\boldsymbol{x}, y))$, minimizing the augmented supervised loss $\mathcal{L}_{\text{SL}}(\pi(\boldsymbol{x}+\delta(\boldsymbol{x}, y), y; f)$ implicitly minimizes the contrastive loss $\mathcal{L}_{\text{CL}}(\boldsymbol{x}+\delta(\boldsymbol{x}, y); g)$ which appears in contrastive error minimization Equ. 3. In other words, supervised error-minimizing noises with enhanced data augmentations can partially replace the functionality of contrastive error-minimizing noises to create shortcuts for contrastive learning.

In Figure 3a, we verify this insight by gradually increasing the augmentation strength $s$ in the supervised error-minimizing framework according to Algorithm 1. The SimCLR accuracy prominently

---

**Algorithm 1** Augmented Unlearnable Examples (AUE)

---

**Require:** Augmentation strength $s$ and a corresponding augmentation distribution $\mu_s$. A labeled training set $\mathcal{D}_c = \{(\boldsymbol{x}_i, y_i)\}_{i=1}^r$. An initialized classifier $f_\theta$. Total epochs $T$, model update iterations $T_\theta$, poisons update iterations $T_\delta$, and perturbation steps $T_p$. Learning rate $\alpha_\theta, \alpha_\delta$.
**Ensure:** Poisons $\{\boldsymbol{\delta}_i\}_{i=1}^r$
$\quad \boldsymbol{\delta}_i \leftarrow 0, i = 1, 2, \cdots, r$              ▷ Initialize poisons
$\quad$ **for** $t = 1, \cdots, T$ **do**
$\quad\quad$ **for** $t_\theta = 1, \cdots, T_\theta$ **do**            ▷ Update the model
$\quad\quad\quad$ Sample a data batch $\{(\boldsymbol{x}_{l_j}, y_{l_j})\}_{j=1}^m$ and an augmentation batch $\{\pi_{l_j} \sim \mu_s\}_{j=1}^m$
$\quad\quad\quad \theta \leftarrow \theta - \frac{\alpha_\theta}{m} \cdot \sum_{j=1}^m \nabla_\theta \mathcal{L}_{\mathrm{SL}}(\pi_{l_j}(\boldsymbol{x}_{l_j} + \boldsymbol{\delta}_{l_j}), y_{l_j}; f_\theta)$
$\quad\quad$ **for** $t_\delta = 1, \cdots, T_\delta$ **do**             ▷ Update poisons
$\quad\quad\quad$ Sample a batch of data $\{(\boldsymbol{x}_{l_j}, y_{l_j})\}_{j=1}^m$
$\quad\quad\quad$ **for** $t_p = 1, \cdots, T_p$ **do**
$\quad\quad\quad\quad$ Sample a augmentation batch $\{\pi_{l_j} \sim \mu\}_{j=1}^m$
$\quad\quad\quad\quad \boldsymbol{\delta}_{l_j} \leftarrow \mathrm{Clip}_\epsilon \left( \boldsymbol{\delta}_{l_j} - \alpha_\delta \cdot \mathrm{sign}(\nabla_{\boldsymbol{\delta}_{l_j}} \mathcal{L}_{\mathrm{SL}}(\pi_{l_j}(\boldsymbol{x}_{l_j} + \boldsymbol{\delta}_{l_j}), y_{l_j}; f_\theta))\right), j = 1, 2, \cdots, m$

---

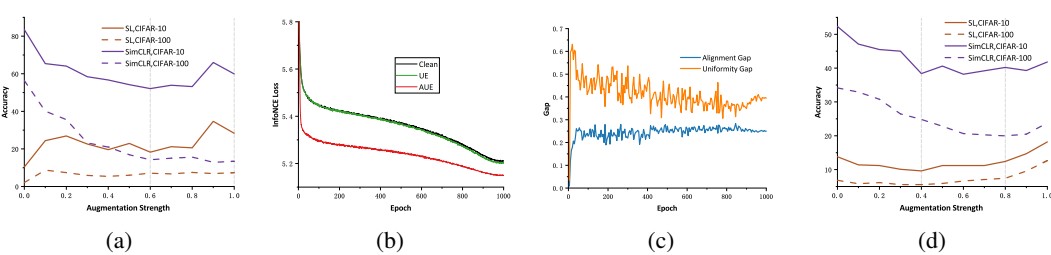

(a)       (b)       (c)       (d)

Figure 3: (a) Influence of augmentations in AUE. (b) Contrastive losses during SimCLR training under UE and AUE attacks. (c) Alignment and uniformity gaps during the SimCLR training on CIFAR-10 poisoned by AUE attack. (d) Influence of augmentations in targeted AAP.

decreases as the strength grows, while the supervised learning accuracy slightly increases. Compared to UE, our AUE attacks largely improve contrastive unlearnability while keeping similar supervised unlearnability. On CIFAR-10, too strong strengths might compromise the unlearnability. Thus, we generate our augmented unlearnable example (AUE) attacks taking $s = 0.6$ for CIFAR-10, and $s = 1.0$ for CIFAR-100. On CIFAR-10, AUE improves $5.7\%$ and $36.6\%$ for supervised unlearnability and contrastive (SimCLR) unlearnability respectively. In Figure 3b, AUE noises largely reduce the contrastive loss during SimCLR training compared to UE noises. It verifies that supervised error minimization with enhanced augmentations mimics contrastive error minimization to some extent. In Figure 3c , we investigate the alignment and uniformity gaps and discuss more about the poisoned training process in Section 6.3. The final gaps of AUE are $\mathcal{AG} = 0.25, \mathcal{UG} = 0.39$ while those of UE are $\mathcal{AG} = 0.05, \mathcal{UG} = 0.03$. Enhanced data augmentations help supervised error-minimizing noises create shortcuts for CL.

## 5.3 AUGMENTED ADVERSARIAL POISONING (AAP)

Adversarial poisoning attacks in Equ. 4 first train a supervised reference model, and then generate adversarial examples on the reference model. For targeted AP, while reference model training uses standard supervised loss, the loss for noise generation translates class labels by $K$ such that generated poisons contain non-robust features that are related to the shifted labels. When we generate adversarial poisoning with enhanced data augmentations $\pi \sim \mu$, minimizing $\mathbb{E}\left[\mathcal{L}_{\mathrm{SL}}(\pi(\boldsymbol{x}), y; f)\right]$ with respect to $f$ mimics updating a reference model with contrastive training. Then, minimizing $\mathcal{L}_{\mathrm{SL}}(\pi(\boldsymbol{x} + \delta(\boldsymbol{x}, y)), y + K; f^*)$ with respect to $\delta$ updates poisons to deceive a contrastive-like reference model $f^*$. For untargeted AP, stronger data augmentations play a similar role. As a consequence, the resulting poisons can learn more about how to confound contrastive learning algorithms. According to Algorithm 2, we gradually increase the augmentation strength $s$ in both inner and outer optimization from $0.0$ to $1.0$ and set the label translation $K = 1$. In Figure 3d, the SimCLR accuracy decreases with the strength, while the supervised learning accuracy slightly

---

**Algorithm 2** Augmented Adversarial Poisoning (AAP)

---

**Require:** Similar to the setting in Algorithm 1.
**Ensure:** Poisons $\{\boldsymbol{\delta}_i\}_{i=1}^r$

    $\boldsymbol{\delta}_i \leftarrow 0, i = 1, 2, \cdots, r$                                         ▷ Initialize poisons
    **for** $t = 1, \cdots, T$ **do**                                     ▷ Reference model
        **for** $t_\theta = 1, \cdots, T_\theta$ **do**
            Sample a data batch $\{(\boldsymbol{x}_{l_j}, y_{l_j})\}_{j=1}^m$ and an augmentation batch $\{\pi_{l_j} \sim \mu_s\}_{j=1}^m$
            $\theta \leftarrow \theta - \frac{\alpha_\theta}{m} \cdot \sum_{j=1}^m \nabla_\theta \mathcal{L}_{\text{SL}}(\pi_{l_j}(\boldsymbol{x}_{l_j}), y_{l_j}; f_\theta)$
    **for** $i = 1, \cdots, r$ **do**                                 ▷ Adversarial examples
        **for** $t_p = 1, \cdots, T_p$ **do**
            Sample $\pi_i \sim \mu_s$
            $\boldsymbol{\delta}_i \leftarrow \text{Clip}_\epsilon\big(\boldsymbol{\delta}_i + \alpha_\delta \cdot \text{sign}(\nabla_{\boldsymbol{\delta}_i}\mathcal{L}_{\text{SL}}(\pi_i(\boldsymbol{x}_i + \boldsymbol{\delta}_i), y_i; f_\theta))\big)$     ▷ Untargeted AAP
            $\boldsymbol{\delta}_i \leftarrow \text{Clip}_\epsilon\big(\boldsymbol{\delta}_i - \alpha_\delta \cdot \text{sign}(\nabla_{\boldsymbol{\delta}_i}\mathcal{L}_{\text{SL}}(\pi_i(\boldsymbol{x}_i + \boldsymbol{\delta}_i), y_i + 1; f_\theta))\big)$     ▷ Targeted AAP

---

increases. Proper augmentation strengths improve the contrastive unlearnability but too large $s$ might introduce difficulty in poison generation and harm the supervised unlearnability. We select $s = 0.4$ for CIFAR-10 and $s = 0.8$ for CIFAR-100. Our T-AAP attacks further improve the contrastive unlearnability of Targeted AP by $9.3\%$ on CIFAR-10 and $5.5\%$ on CIFAR-100 for SimCLR.

## 6 EXPERIMENTS

### 6.1 SETUP

Poisons are generated on CIFAR-10/100, Tiny-ImageNet, modified Mini-ImageNet, and ImageNet-100. ResNet-18 He et al. (2016) is used for poison generation and evaluation if not otherwise stated. Our threat model considers the worst-case unlearnability across supervised and contrastive algorithms. A standard supervised learning algorithm and four contrastive learning algorithms including SimCLR (Chen et al., 2020a), MoCo v2 (Chen et al., 2020b), BYOL (Grill et al., 2020) and SimSiam (Chen & He, 2021) are employed to evaluate the attack performance of availability attacks. We adopt a linear probing stage on the poisoned data. We adopt AP, SEP-FA-VR, CP, and TUE as baselines for the worst-case unlearnability. T-AP and T-AAP are targeted attacks and UT-AP and UT-AAP are untargeted. Our AUE and AAP train reference models from scratch rather than using pre-trained weights. For CP and TUE attacks, we specify algorithms they used for noise generation, for example, CP-SimCLR. Moreover, we also evaluate class-wise CP attacks which are denoted by C-CP. Detailed settings for evaluations and our proposed attacks are shown in Appendix A.

### 6.2 WORST-CASE UNLEARNABILITY

In Table 2, compared to existing availability attacks, our proposed attacks achieve state-of-the-art worst-case unlearnability across five evaluation algorithms on CIFAR-10 and CIFAR-100. Since the generation of untargeted adversarial poisoning is unstable (Fowl et al., 2021), we generate UT-AAP only on CIFAR-10. A contrastive learning-based attack such as CP and TUE relies on a specific contrastive algorithm and possibly loses the unlearnability against other supervised or contrastive algorithms. For example, on CIFAR-100, sample-wise CP attacks fail to deal with supervised learning and TUE-SimCLR performs poorly against BYOL. However, AUE and AAP are based on supervised training but show more stable unlearnability against different contrastive algorithms.

In Table 2 and 3, we also evaluate the attack performance of our attacks on high-resolution datasets including Mini-ImageNet, Tiny-ImageNet, and ImageNet-100, where we set augmentation strength $s = 1.0$ for AUE and $s = 0.8$ for AAP as the same as on CIFAR-100. In general, T-AAP improves the worst-case un-

Table 3: Performance on ImageNet-100.

| Attack | SL | SimCLR | MoCo | BYOL | SimSiam |
|---|---|---|---|---|---|
| Clean | 77.8 | 61.8 | 61.8 | 62.2 | 65.8 |
| AUE | **5.1** | **5.2** | **6.2** | **7.5** | **4.7** |
| T-AAP | 14.4 | 20.3 | 14.5 | 24.8 | 16.6 |

learnability compared to T-AP, and AUE outperforms T-AAP and TUE-MoCo in the worst-case unlearnability.

Table 2: Worst-case unlearnability of availability attacks across supervised and contrastive algorithms.

| Dataset | Attack | SL | Contrastive Learning | | | | Worst |
| | | | SimCLR | MoCo | BYOL | SimSiam | |
|---|---|---|---|---|---|---|---|
| CIFAR-10 | Clean | 95.5 | 91.3 | 91.5 | 92.3 | 90.7 | 95.5 |
| | T-AP | 9.5 | 48.4 | 53.8 | 53.0 | 51.1 | 53.8 |
| | UT-AP | 9.6 | 41.5 | 31.5 | 44.0 | 42.8 | 44.0 |
| | SEP-FA-VR | **2.3** | 37.3 | 35.8 | 42.8 | 36.7 | 42.8 |
| | CP-SimCLR | 94.5 | 38.7 | 69.3 | 79.5 | **29.2** | 94.5 |
| | CP-MoCo | 94.5 | 53.7 | 47.9 | 56.8 | 47.1 | 94.5 |
| | CP-BYOL | 11.0 | 39.3 | 32.7 | 41.8 | 37.9 | 41.8 |
| | C-CP-SimCLR | 9.4 | 50.1 | 50.6 | 47.7 | 57.7 | 57.7 |
| | C-CP-MoCo | 10.0 | 50.8 | 51.6 | 47.0 | 51.2 | 51.6 |
| | C-CP-BYOL | 10.7 | 44.5 | 40.2 | 43.4 | 41.4 | 44.5 |
| | TUE-SimCLR | 10.6 | 48.1 | 71.2 | 79.5 | 39.0 | 79.5 |
| | TUE-MoCo | 10.1 | 57.2 | 51.6 | 60.1 | 58.5 | 60.1 |
| | TUE-SimSiam | 9.9 | 82.5 | 80.7 | 84.3 | 81.8 | 84.3 |
| | AUE (ours) | 18.9 | 52.4 | 57.0 | 58.2 | 34.5 | 58.6 |
| | T-AAP (ours) | 9.2 | 39.1 | 40.4 | 43.3 | 42.1 | 43.3 |
| | UT-AAP (ours) | 29.7 | **32.3** | **23.2** | **35.5** | 34.1 | **35.5** |
| CIFAR-100 | Clean | 77.4 | 63.9 | 67.9 | 63.7 | 64.4 | 77.4 |
| | T-AP | 3.2 | 25.6 | 26.6 | 26.1 | 28.8 | 28.8 |
| | UT-AP | 42.7 | 11.1 | **9.8** | **10.1** | 14.0 | 42.7 |
| | SEP-FA-VR | 2.4 | 25.2 | 25.9 | 26.6 | 28.4 | 28.4 |
| | CP-SimCLR | 74.7 | **10.5** | 30.7 | 22.6 | **7.7** | 74.7 |
| | CP-MoCo | 74.4 | 15.2 | 13.4 | 16.4 | 14.1 | 74.4 |
| | CP-BYOL | 74.7 | 29.7 | 35.5 | 35.7 | 29.5 | 74.7 |
| | C-CP-SimCLR | 1.0 | 14.9 | 25.3 | 24.4 | 27.6 | 27.6 |
| | C-CP-MoCo | 1.0 | 22.9 | 20.4 | 25.8 | 26.1 | 26.1 |
| | C-CP-BYOL | 1.0 | 25.2 | 25.1 | 23.2 | 28.8 | 28.8 |
| | TUE-SimCLR | **1.0** | 16.9 | 36.7 | 40.6 | 7.8 | 40.6 |
| | TUE-MoCo | 1.0 | 19.9 | 19.6 | 22.3 | 18.6 | 22.3 |
| | TUE-SimSiam | 1.1 | 33.9 | 31.0 | 40.9 | 10.3 | 40.9 |
| | AUE (ours) | 6.9 | 13.6 | 19.0 | 19.2 | 11.9 | **19.2** |
| | T-AAP (ours) | 7.3 | 20.1 | 18.6 | 21.1 | 21.3 | 21.3 |
| Mini-ImageNet | Clean | 66.2 | 55.3 | 57.6 | 48.7 | 54.5 | 66.2 |
| | T-AP | 11.5 | 48.9 | 50.1 | 44.0 | 48.5 | 50.1 |
| | TUE-MoCo | 9.8 | 46.2 | 48.4 | 43.1 | 46.9 | 48.4 |
| | AUE (ours) | **8.7** | **15.0** | **20.4** | **14.5** | **18.2** | **20.1** |
| | T-AAP (ours) | 29.8 | 43.8 | 41.9 | 40.2 | 41.8 | 43.8 |
| Tiny-ImageNet | Clean | 53.5 | 39.6 | 43.3 | 33.9 | 42.4 | 53.5 |
| | T-AP | 11.3 | 32.8 | 34.7 | 27.2 | 34.5 | 34.7 |
| | TUE-MoCo | **5.5** | 20.9 | 24.9 | 20.3 | 25.0 | 25.0 |
| | AUE (ours) | 7.1 | **10.8** | **11.7** | **9.6** | **11.6** | **11.7** |
| | T-AAP (ours) | 18.7 | 28.4 | 27.6 | 25.2 | 28.2 | 28.4 |

## 6.3 POISONED TRAINING PROCESS

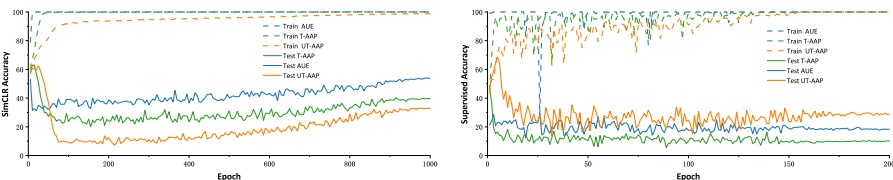

Figure 4: Poisoned SimCLR (every 5 epochs) and SL (every epoch) on CIFAR-10.

In Figure 4, we evaluate the training and test accuracy during SL and CL training. In very early epochs where the training underfits the corresponding objective, checkpoints possibly process weak unlearnability for both SL and CL since supervised and contrastive shortcuts have not been learned well. Early stop training can mitigate the unlearnability. As shown in Figure 3c for AUE, after a few epochs, the shortcuts have been established and thus the SimCLR accuracy suddenly decreases. Moreover, in the middle and later stages of training, the slow increase in CL accuracy aligns with the overall trend of gradually decreasing uniformity gap and relatively stable alignment gap.

## 6.4 DECOUPLING THE AUGMENTAITONS

In previous settings of AUE and AAP, we control the strength of RandomResizeCrop, RandomColor-Jitter, and RancomGrayscale through a single strength hyperparameter $s$ in the poison generation, as shown in Code 1. In Table 4, we decouple the strength hyperparameters for these three random transforms and evaluate the resulting attacks by SimCLR. Different factors show different influences on the contrastive unlearnability for AUE and T-AAP. However, adjusting these three factors together outperforms other options in conclusion.

Table 4: SimCLR evaluation of attacks generated with decoupled strength parameters on CIFAR-10. By default, $s = 0.6$ for AUE and $s = 0.4$ for T-AAP. For example, 0-0-$s$ means that RandomResize-Crop strength is 0, RandomColorJitter strength is 0, and RancomGrayscale strength is $s$.

|       | 0-0-0 | 0-0-$s$ | 0-$s$-0 | $s$-0-0 | 0-$s$-$s$ | $s$-0-$s$ | $s$-$s$-0 | $s$-$s$-$s$ |
|-------|-------|---------|---------|---------|-----------|-----------|-----------|-------------|
| AUE   | 83.5  | 58.7    | 79.4    | 88.7    | 60.8      | 56.2      | 87.7      | **52.4**    |
| T-AAP | 52.3  | 52.0    | 52.9    | 44.9    | 51.4      | 42.2      | 44.8      | **39.1**    |

## 6.5 DEFENSES

In Table 5, we investigate existing defenses against AUE and AAP attacks on CIFAR-10 for SL and CL respectively. For supervised learning, we adopt adversarial training AT (Madry et al., 2018), ISS variants (Liu et al., 2023), UEraser variants (Qin et al., 2023), and AVATAR (Dolatabadi et al., 2023). For contrastive learning, we adopt contrastive adversarial training AdvCL (Kim et al., 2020), and Sim-CLR models with Cutout (length 8) (DeVries & Taylor, 2017), Random noise (variance $8/255$), and Gaussian Blur (kernel size 3).

For SL, AVATAR and ISS-JPEG achieve better defense performance than AT. UEraser variants are not very effective for AUE. For CL, AdvCL and Gaussian Blur have similar effects against T-AAP and UT-AAP. And AdvCL is the most effective defense against AUE.

Table 5: Defenses against proposed attacks on CIFAR-10 for SL and CL.

| Defense       | AUE      | T-AAP    | UT-AAP   |
|---------------|----------|----------|----------|
| UEraser       | **63.2** | 64.7     | 81.8     |
| UEraser-Lite  | **60.6** | 66.8     | 82.2     |
| UEraser-Max   | **72.8** | 79.5     | 85.8     |
| ISS           | 82.6     | 82.3     | **81.4** |
| ISS-Grayscale | 18.2     | **9.1**  | 23.8     |
| ISS-JPEG      | 84.9     | 84.3     | **84.0** |
| AVATAR        | **85.0** | 88.0     | 86.6     |
| AT-8/255      | 83.8     | 81.6     | **79.6** |
| Cutout        | 51.8     | 37.9     | **31.8** |
| Random Noise  | 60.5     | 62.4     | **48.0** |
| Gaussian Blur | **69.1** | 76.7     | 78.9     |
| AdvCL-8/255   | 80.9     | 78.4     | **77.5** |

## 7 CONCLUSION

Since contrastive learning brings new challenges to protect data using availability attacks, we explore availability attacks that have worst-case unlearnability across supervised and contrastive algorithms. To our knowledge, the label information is necessary for the poisoning generation to acquire stable worst-case unlearnability. For example, TUE uses a class-wise separability discriminant loss which contains label information to achieve supervised unlearnability and class-wise CP requires label information as well. Thus, we start from basic supervised poisoning methods and find that enhanced data augmentations can boost their contrastive unlearnability. Our proposed AUE and AAP attacks achieve state-of-the-art worst-case unlearnability across multiple supervised and contrastive algorithms.

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

## A EXPERIMENT DETAILS

### A.1 DATASETS AND NETWORKS

**CIFAR.** CIFAR-10/CIFAR-100 Krizhevsky et al. (2009) consists of 50000 training images and 10000 test images in 10/100 classes. All images are $32 \times 32$ colored ones.

**Tiny-ImageNet.** Tiny-Imagenet classification challenge (Le & Yang, 2015) is similar to the classification challenge in the full ImageNet ILSVRC (Russakovsky et al., 2015). It contains 200 classes. The training has 500 images for each class and the test set has 100 images for each class. All images are $64 \times 64$ colored ones.

**Mini-ImageNet.** Mini-ImageNet dataset was originally designed for few-shot learning (Vinyals et al., 2016). We modify it for a classification task. The modified dataset contains 100 classes. The training set has 500 images for each class. The test set has 100 images for each class. All images are $84 \times 84$ colored ones.

**ImageNet-100.** ImageNet-100 is a subset of ImageNet-1k Dataset from ImageNet Large Scale Visual Recognition Challenge 2012 (Russakovsky et al., 2015). It contains 100 random classes. The training set has 130,000 images. The test set has 5,000 images.

**ResNet.** On CIFAR-10/CIFAR-100, we set the kernel size of the first convolutional layer to 3 and removed the following max-pooling layer. On other datasets, we do not modify the models.

### A.2 AUGMENTATIONS

Code Listing 1: Different data augmentations used in supervised learning and contrastive learning on CIFAR-10/100 datasets. The intensity of contrastive augmentations can be adjusted via strength $s$.

```
# Supervised augmentations
Compose([RandomCrop(size=32, padding=4), RandomHorizontalFlip(p=0.5),
         ToTensor()])
# Contrastive augmentations
s = 1.0  # Strength is 1.0 by default for contrastive learning.
Compose([RandomResizedCrop(size=32, scale=(1-0.9*s, 1.0)),
         RandomHorizontalFlip(p=0.5),
         RandomApply([ColorJitter(brightness=0.4*s, contrast=0.4*s,
                                  saturation=0.4*s, hue=0.1*s)], p=0.8*s),
         RandomGrayscale(p=0.2*s), ToTensor()])
```

### A.3 DETAILS OF AUE AND AAP

We leverage differentiable augmentation modules in Konia[1] (Riba et al., 2020) which is a differentiable computer vision library for PyTorch. The contrastive augmentations for Tiny/Mini-ImageNet, and ImageNet-100 are similar to those for CIFAR-10/100 in Code 1 but only adapt the image size.

**AUE.** We train the reference model for $T = 60$ epochs with SGD optimizer and cosine annealing learning rate scheduler. The batch size of training data is 128. The initial learning rate $\alpha_\theta$ is 0.1, weight decay is $10^{-4}$ and momentum is 0.9. In each epoch, we update the model for $T_\theta = 391$ iterations and update poisons for $T_\delta = 391$ iterations. For ImageNet-100, we set $T_\theta = T_\delta = 1016$. The PGD process for noise generation takes $T_p = 5$ steps with step size $\alpha_\delta = 0.8/255$.

The augmentation strength $s = 0.6$ for CIFAR-10 and $s = 1.0$ for CIFAR-100, Mini-ImageNet, Tiny-ImageNet, and ImageNet-100.

**AAP.** We train the reference model for $T = 40$ epochs, and the initial learning rate $\alpha_\theta$ is 0.5. The PGD process for noise generation takes $T_p = 250$ steps with step size $\alpha_\delta = 0.08/255$. Other settings are the same as AUE. The label translation is $K = 1$.

The augmentation strength $s = 0.4$ for CIFAR-10 and $s = 0.8$ for CIFAR-100, Mini-ImageNet, Tiny-ImageNet, and ImageNet-100.

---

[1]https://github.com/kornia/kornia

### A.4 EVALUATION ALGORITHMS

The setup for SimCLR, MoCo v2, BYOL, and SimSiam are shown in Table 6. The 100-epoch linear probing stage uses an SGD optimizer and a scheduler that decays 0.2 at 60, 75, and 90 epochs. The probing learning rate is 1.0 for SimCLR, MoCo v2, BYOL, and 5.0 for SimSiam on CIFAR-10/100, Tiny/Mini-ImageNet.

On ImageNet-100, the unsupervised contrastive learning optimizes 200 epochs and the linear probing uses a learning rate of 10.0. Other settings are the same as other datasets.

For supervised learning, we augment the training data by RandomHorizontalFlip and RandomCrop with padding size $l/8$ on CIFAR-10/100 and Tiny/Mini-ImageNet. $l$ is the image size. On ImageNet-100, we augment using RandomResizedCrop and RandomHorizontalFlip.

Table 6: Details of supervised and contrastive evaluations.

|                  | SL     | SimCLR  | MoCo v2 | BYOL    | SimSiam    |
|------------------|--------|---------|---------|---------|------------|
| Batch size       | 512    | 512     | 512     | 512     | 512        |
| Epochs           | 200    | 1000    | 1000    | 1000    | 1000       |
| Loss function    | CE     | InfoNCE | InfoNCE | MSE     | Similarity |
| Optimizer        | SGD    | SGD     | SGD     | SGD     | SGD        |
| Learning rate    | 0.5    | 0.5     | 0.3     | 1.0     | 0.1        |
| Weight decay     | 1e-4   | 1e-4    | 1e-4    | 1e-4    | 1e-4       |
| Momentum         | 0.9    | 0.9     | 0.9     | 0.9     | 0.9        |
| Scheduler        | Cosine | Cosine  | Cosine  | Cosine  | Cosine     |
| Warmup           | 10     | 10      | 10      | 10      | 10         |
| Temperature      | -      | 0.5     | 0.2     | -       | -          |
| Encoder momentum | -      | -       | 0.99    | 0.999   | -          |

## B ADDITIONAL EXPERIMENTS

### B.1 COMPUTATION CONSUMPTION

We report the time consumption of generating AUE and AAP attacks. For CIFAR-10/100, Tiny/Mini-Imagenet, experiments are conducted using a single NVIDIA GeForce RTX 3090 GPU. For Imagenet-100, experiments are conducted using a single NVIDIA A800 GPU.

On CIFAR-10/100, AUE/AAP costs around 2.7/2.2 hours. On Mini-ImageNet, AUE/AAP costs around 2.5/2 hours. On Tiny-ImageNet, AUE/AAP costs around 2.5/3.8 hours. On ImageNet-100, AUE/AAP costs around 12/10 hours.

In comparison, on CIFAR-10/100 and using the same device, CP-SimCLR costs around 48 hours, and TUE-MoCo costs around 8.5 hours to generate poisons. Our supervised poisoning attacks are much more efficient than contrastive poisoning attacks.

### B.2 VISUALIZATION

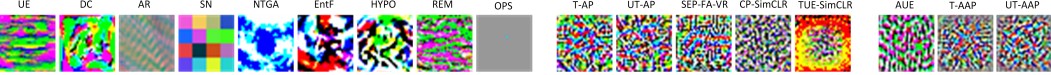

Figure 5: Noise images of availability attacks on CIFAR-10.

In Figure 5, we present noise images of availability attacks on CIFAR-10. Compared to those unable to deceive contrastive learning, attacks that have contrastive unlearnability possess more complicated and high-frequency features. Considering contrastive augmentations including grayscale can eliminate low-frequency shortcuts Liu et al. (2023), noises at least need to come through these augmentations to be effective for contrastive learning. Our proposed attacks generate noises that are adaptive to these augmentations during the generation process.

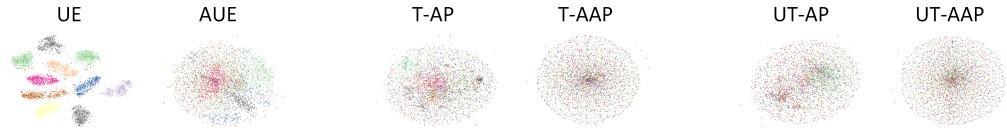

Figure 6: T-SNE visualization of noises.

In Figure 6, we compare the t-SNEs of our proposed noises and their basic counterparts. While UE noises cluster, our AUE noises are more dispersed. Considering that AUE has even stronger supervised unlearnability than UE, enhanced data augmentations help generate more complex unlearnable features that work as shortcuts for supervised learning and contrastive learning together. Compared to AP noises, our AAP noises are more dispersed as well.

## B.3 Transferability across networks.

We generate AUE and AAP using ResNet-18 and test them on ResNet-50, VGG-19 (Simonyan & Zisserman, 2015), DenseNet-121 (Huang et al., 2017), and MobileNet v2 (Howard et al., 2017; Sandler et al., 2018). In Table 7, both supervised unlearnability and contrastive unlearnability of AUE and AAP can transfer across these architectures. Moreover, the relative attack performance is preserved: T-AAP is consistently best for supervised learning and UT-AAP is consistently best for contrastive learning.

Table 7: Transferability across network architectures on CIFAR-10.

| Network | AUE | T-AAP | UT-AAP |
|---|---|---|---|
| SL-ResNet-50 | 16.4 | **8.9** | 33.2 |
| SL-VGG-19 | 23.2 | **10.7** | 43.5 |
| SL-DenseNet-121 | 19.5 | **10.4** | 37.5 |
| SL-MobileNet v2 | 17.2 | **12.1** | 27.8 |
| SimCLR-ResNet-50 | 53.4 | 41.5 | **38.4** |
| SimCLR-VGG-19 | 48.2 | 41.7 | **18.0** |
| SimCLR-DenseNet-121 | 50.5 | 35.3 | **31.3** |
| SimCLR-MobileNet v2 | 41.4 | 29.8 | **19.9** |

## B.4 Poisoning ratio.

Since basic approaches, UE and AP need to poison almost the whole dataset, we evaluate the influence of the poisoning ratio on the attack performance of AUE and AAP. Table 8 illustrates that clean data dilution can largely destroy unlearnability. Supervised unlearnability is more sensitive than contrastive unlearnability and AUE is more robust to poisoning ratio than AAP.

Table 8: The influence of poisoning ratio on CIFAR-10.

| Ratio | AUE | T-AAP | UT-AAP |
|---|---|---|---|
| SL-95% | **75.6** | 82.1 | 84.2 |
| SL-90% | **82.2** | 86.6 | 89.2 |
| SL-80% | **87.6** | 89.8 | 91.2 |
| SimCLR-95% | **69.7** | 76.8 | 74.2 |
| SimCLR-90% | **74.5** | 82.1 | 79.9 |
| SimCLR-80% | **79.7** | 85.5 | 83.9 |

## B.5  POISONING BUDGET.

In Table 9, we investigate the attack performance with different poisoning budgets. Contrastive unlearnability requires a larger budget than supervised unlearnability.

Table 9: Influence of poisoning budget on CIFAR-10.

|  | AUE | T-AAP | UT-AAP |
|---|---|---|---|
| SL-2/255 | 34.5 | 50.7 | 75.6 |
| SL-4/255 | 28.5 | 19.7 | 58.5 |
| SL-6/255 | **26.8** | **12.3** | **44.2** |
| SimCLR-2/255 | 84.8 | 87.0 | 87.1 |
| SimCLR-4/255 | 70.1 | 66.6 | 59.8 |
| SimCLR-6/255 | **59.4** | **51.1** | **43.0** |

## B.6  STRENGTH AND GAPS

On CIFAR-10, we gradually increase the augmentation strength from 0 to the default setting, i.e. $s = 0.6$ in the generation of AUE attacks and evaluate the alignment gaps, uniformity gaps, and the SimCLR Accuracy in Table 10. In this case, the larger the gaps, the lower the accuracy of SimCLR.

Table 10: Alignment and uniformity gaps in AUE with different strengths on CIFAR-10.

| Strength | Alignment Gap | Uniformity Gap | SimCLR Accuracy |
|---|---|---|---|
| $s = 0.0$ | 0.14 | 0.07 | 83.5 |
| $s = 0.2$ | 0.21 | 0.24 | 64.1 |
| $s = 0.4$ | 0.25 | 0.28 | 56.7 |
| $s = 0.6$ | 0.25 | 0.39 | 52.4 |

## B.7  DISCUSSION OF CLEAN LINEAR PROBING.

While our threat model linear probes on poisoned data, He et al. (2022); Ren et al. (2022) use clean data for linear probing instead. In Figure 7, we compare the final classification performance of Sim-CLR models in these two settings. Feature extractors are trained on poisoned data and we focus on the classification performance after linear probing on clean and poisoned data respectively. While CP and TUE obtain similar attack performance in both cases, clean linear probing can mitigate supervised training-based attacks including AP, SEP-FA-VR, AAP, and AUE. For supervised poisoning, the dissimilarities between clean features and poisoned features hinder a classifier head obtained by poisoned linear probing in generalizing to clean features. However, clean features still contain some useful information and can derive another classifier head to perform classification. On the other hand, contrastive error-minimizing noises

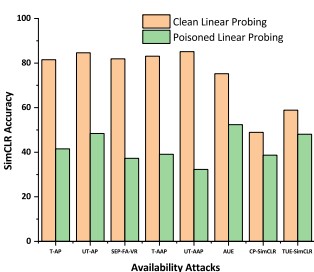

Figure 7: Clean and poisoned linear probing on CIFAR-10.

confuse the feature extractor directly such that even clean data fail to activate useful features for classification. Given a responsible data publisher who protects data using availability attacks before release, an unauthorized data collector has no access to unprocessed data for clean linear probing. Thus, it is sufficient to achieve contrastive unlearnability with poisoned linear probing in real scenarios.

## C  PROOF OF THEOREM 5.1

### C.1  LEMMAS

We use notations in Theorem 5.1.

**Lemma C.1.** *For any $\boldsymbol{z} \in \mathbb{R}^n$,*

$$\sigma_n ||\boldsymbol{z}|| \leq ||W\boldsymbol{z}|| \leq \sigma_1 ||\boldsymbol{z}||.$$

*Proof.* Denote $\tilde{\boldsymbol{z}} = (\tilde{z}_1, \cdots, \tilde{z}_n)^\top = V\boldsymbol{z}$. Since orthogonal matrices preserve the norm,

$$||W\boldsymbol{z}|| = ||U\Sigma V\boldsymbol{z}|| = ||\Sigma\tilde{\boldsymbol{z}}|| = \sqrt{\sum_{i=1}^{n} \sigma_i^2 \tilde{z}_i^2},$$

$$\sigma_n ||\boldsymbol{z}|| = \sigma_n ||\tilde{\boldsymbol{z}}|| \leq \sqrt{\sum_{i=1}^{n} \sigma_i^2 \tilde{z}_i^2} \leq \sigma_1 ||\tilde{\boldsymbol{z}}|| = \sigma_1 ||\boldsymbol{z}||.$$

$\square$

**Lemma C.2.** *If $\mathcal{E}_{\mathrm{SL}} \leq \epsilon$, then with probability at least $1 - \sqrt{\epsilon}$*

$$||h \circ g(\pi(\boldsymbol{x})) - \boldsymbol{e}_y|| < \sqrt{n\sqrt{\epsilon}},$$

*where $(\boldsymbol{x}, y) \sim \mathcal{D}, \pi \sim \mu$.*

*Proof.* As

$$\mathcal{E}_{\mathrm{SL}} = \mathop{\mathbb{E}}_{\substack{(\boldsymbol{x},y) \sim \mathcal{D} \\ \pi \sim \mu}} \left[ \frac{1}{n} ||h \circ g(\pi(\boldsymbol{x})) - \boldsymbol{e}_y||^2 \right],$$

by Markov's inequality, it has

$$\Pr(\frac{1}{n} ||h \circ g(\pi(\boldsymbol{x})) - \boldsymbol{e}_y||^2 \geq \sqrt{\epsilon}) \leq \sqrt{\epsilon}.$$

$\square$

**Lemma C.3.** *If $\mathcal{E}_{\mathrm{SL}} \leq \epsilon$, then with probability at least $1 - 2\sqrt{\epsilon}$*

$$g(\pi(\boldsymbol{x}))^\top g(\tau(\boldsymbol{x})) > 1 - \frac{2n\sqrt{\epsilon}}{\sigma_n},$$

*where $\boldsymbol{x} \sim \mathcal{D}_{\boldsymbol{x}}, \pi, \tau \sim \mu$.*

*Proof.* By Lemma C.2, with probability at least $1 - 2\sqrt{\epsilon}$,

$$||h \circ g(\pi(\boldsymbol{x})) - \boldsymbol{e}_y|| < \sqrt{n\sqrt{\epsilon}} \quad \text{and} \quad ||h \circ g(\tau(\boldsymbol{x})) - \boldsymbol{e}_y|| < \sqrt{n\sqrt{\epsilon}}.$$

By the triangle inequality,

$$||h \circ g(\pi(\boldsymbol{x})) - h \circ g(\tau(\boldsymbol{x}))|| < 2\sqrt{n\sqrt{\epsilon}}$$

Since $g$ is normalized, by Lemma C.1 we have

$$g(\pi(\boldsymbol{x}))^\top g(\tau(\boldsymbol{x})) = 1 - \frac{1}{2} ||g(\pi(\boldsymbol{x})) - g(\tau(\boldsymbol{x}))||^2$$

$$\geq 1 - \frac{1}{2\sigma_n^2} ||h \circ g(\pi(\boldsymbol{x})) - h \circ g(\tau(\boldsymbol{x}))||^2$$

$$> 1 - \frac{2n\sqrt{\epsilon}}{\sigma_n}.$$

$\square$

**Lemma C.4.** *Assume $\mathcal{D}$ is a balanced dataset. If $\mathcal{E}_{\mathrm{SL}} \leq \epsilon$, then with probability at least $1 - 2\sqrt{\epsilon}$, one of the following two conditions holds*

1. *with probability $\frac{n-1}{n}$,*

$$g(\pi(\boldsymbol{x}))^\top g(\tau(\boldsymbol{x}^-)) < 1 - \frac{(1 - \sqrt{2n\sqrt{\epsilon}})^2}{\sigma_1^2};$$

2. *with probability $\frac{1}{n}$,*

$$g(\pi(\boldsymbol{x}))^\top g(\tau(\boldsymbol{x}^-)) \leq 1.$$

*Proof.* 1. With probability $\frac{n-1}{n}$, for $(\boldsymbol{x}, y), (\boldsymbol{x}^-, y^-) \sim \mathcal{D}$, $y \neq y^-$. By Lemma C.2, with probability at least $1 - 2\sqrt{\epsilon}$,

$$||h \circ g(\pi(\boldsymbol{x})) - \boldsymbol{e}_y|| < \sqrt{n\sqrt{\epsilon}} \quad \text{and} \quad ||h \circ g(\tau(\boldsymbol{x}^-)) - \boldsymbol{e}_{y^-}|| < \sqrt{n\sqrt{\epsilon}}.$$

By the triangle inequality,

$$
\begin{aligned}
||g(\pi(\boldsymbol{x})) - g(\tau(\boldsymbol{x}^-))|| &\geq \frac{1}{\sigma_1}||h \circ g(\pi(\boldsymbol{x})) - h \circ g(\tau(\boldsymbol{x}^-))|| \\
&\geq \frac{1}{\sigma_1}(||\boldsymbol{e}_y - \boldsymbol{e}_{y^-}|| - ||h \circ g(\pi(\boldsymbol{x})) - \boldsymbol{e}_y|| - ||h \circ g(\tau(\boldsymbol{x}^-)) - \boldsymbol{e}_{y^-}||) \\
&> \frac{\sqrt{2} - 2\sqrt{n\sqrt{\epsilon}}}{\sigma_1}.
\end{aligned}
$$

Since $g$ is normalized,

$$
\begin{aligned}
g(\pi(\boldsymbol{x}))^\top g(\tau(\boldsymbol{x}^-)) &= 1 - \frac{1}{2}||g(\pi(\boldsymbol{x})) - g(\tau(\boldsymbol{x}^-))||^2 \\
&< 1 - \frac{(1 - \sqrt{2n\sqrt{\epsilon}})^2}{\sigma_1^2}.
\end{aligned}
$$

2. As we assume $\mathcal{D}$ is a balanced dataset, with probability $\frac{1}{n}$, for $(\boldsymbol{x}, y), (\boldsymbol{x}^-, y^-) \sim \mathcal{D}$, $y = y^-$. Since $g$ is normalized,

$$
\begin{aligned}
g(\pi(\boldsymbol{x}))^\top g(\tau(\boldsymbol{x}^-)) &= 1 - \frac{1}{2}||g(\pi(\boldsymbol{x})) - g(\tau(\boldsymbol{x}^-))||^2 \\
&\leq 1 - \frac{1}{2\sigma_1^2}||h \circ g(\pi(\boldsymbol{x})) - h \circ g(\tau(\boldsymbol{x}^-))||^2 \\
&\leq 1.
\end{aligned}
$$

$\square$

## C.2 PROOF OF THEOREM 5.1

*Proof.* Let $\mathcal{E}_{\text{SL}} = \epsilon$. Combining Lemma C.3 and Lemma C.4, for a sample $\boldsymbol{x}$ and its negative sample $\boldsymbol{x}^-$ i.i.d from $\mathcal{D}_{\boldsymbol{x}}$, and data augmentation method $\pi, \tau, \rho$ i.i.d from $\mu$, with probability at least $1 - 4\sqrt{\mathcal{E}_{\text{SL}}}$, it holds that

$$
\begin{aligned}
\mathcal{L}_{\text{CL}}(x, x^-, \pi, \tau, \rho) &= -\log \frac{e^{g(\pi(\boldsymbol{x}))^\top g(\tau(\boldsymbol{x}))}}{e^{g(\pi(\boldsymbol{x}))^\top g(\tau(\boldsymbol{x}))} + e^{g(\pi(\boldsymbol{x}))^\top g(\rho(\boldsymbol{x}^-))}} \\
&= \log\left(1 + \frac{e^{g(\pi(\boldsymbol{x}))^\top g(\rho(\boldsymbol{x}^-))}}{e^{g(\pi(\boldsymbol{x}))^\top g(\tau(\boldsymbol{x}))}}\right) \\
&< \frac{n-1}{n}\log\left(1 + \frac{1 - \frac{(1-\sqrt{2n\sqrt{\mathcal{E}_{\text{SL}}}})^2}{\sigma_1^2}}{1 - \frac{2n\sqrt{\mathcal{E}_{\text{SL}}}}{\sigma_n}}\right) + \frac{1}{n}\log\left(1 + \frac{1}{1 - \frac{2n\sqrt{\mathcal{E}_{\text{SL}}}}{\sigma_n}}\right).
\end{aligned}
$$

$\square$

