# OpenReview forum: "Availability Attacks Need to Create Shortcuts for Contrastive Learning"
_ICLR.cc/2024/Conference — Submitted to ICLR 2024_

### Official Review · Reviewer_YhBA · 2023-10-19

**Soundness:** 2 fair
**Presentation:** 2 fair
**Contribution:** 2 fair
**Rating:** 3
**Confidence:** 5

**Summary:**

The authors' experiments reveal that most data availability attacks designed for supervised learning become ineffective under contrastive learning training methods. They find that supervised training with enhanced data augmentation in reference models can mimic contrastive learning. Consequently, the authors propose sampling from different data distributions within the data distribution. By employing this contrastive learning-like data augmentation approach for training the substitute models, the generated unlearnable noise can provide protection under both supervised learning and contrastive learning conditions.

**Strengths:**

- The authors quantitatively measure the GAP between the attack effectiveness of supervised learning and contrastive learning methods using two contrastive learning metrics.
- By incorporating the data augmentation techniques of contrastive learning into supervised learning, the authors develop an availability attack method that is effective under both supervised learning and contrastive learning training frameworks.

**Weaknesses:**

- The paper is difficult to read and the organization of the content is not very clear.
- The Cross-Entropy (CE) loss and InfoNCE loss may be essentially similar, and using these two losses to reflect the relationship between the two tasks is not particularly convincing.
- Although the paper emphasizes sampling in data augmentation as being introduced from contrastive learning, it bears a resemblance to the Expectation Over Transformation (EOT) used in the reference paper on REM. EOT also involves sampling from data augmentation, and this technique is commonly used in adversarial settings.
- The assumptions made in the theoretical analysis employ a simple linear network, which presents a significant discrepancy from practical settings.
- The resolution of the Tiny-ImageNet dataset is not down-sampled, and the Mini-ImageNet dataset has a limited number of samples per class.

**Questions:**

- Does the value of the GAP affect accuracy? According to the paper, GAP reflects the difference between clean and poison distributions. Logically, the greater the difference, the better the protection effect should be. However, the results in Table 1 do not seem to support this notion.
- Why does the loss of Alignment in Figure 3(c) first decrease and then increase?

---

> ### Author Response · Authors · 2023-11-16
>
> Thanks for your comments and questions.
>
> We have uploaded a new version of our paper. Based on your suggestions and other reviewers’ feedback, we adjusted the paper structure, improved the presentation, and added more experiments. We summarize these modifications in ‘Author’s comments’.
>
> Besides, we’ll make some clarifications as follows:
> - [CE and InfoNCE]  CE and InfoNCE are typical loss functions for SL and CL respectively. Therefore, a decrease in CE loss signifies optimization of the SL task, while a decrease in InfoNCE loss indicates optimization of the CL task.  By employing enhanced data augmentations, the observation that minimizing CE also leads to a decrease in InfoNCE suggests implicit optimization of the CL task. Building upon this insight, we then analyze the toy model and propose that supervised error minimization with enhanced data augmentations can partially replace the role of contrastive error minimization to enlarge the alignment and uniformity gaps and bring contrastive unlearnability. Moreover, stronger augmentations help adversarial poisoning regarding the CE loss generate poisons to deceive a contrastive-like reference model and thus improve the contrastive unlearnability as well.
> - [Augmentations and EOT] The motivation, technique, and goal of augmentations in our methods are different from EOT. The augmentations used by EOT are standard supervised augmentations including RandomCrop and RandomHorizontalFlip. EOT sampling several augmentations and computes the average gradient over these samples.
> However, we use much stronger contrastive augmentations for both reference model update and perturbation update to equip the attack with contrastive unlearnability. In addition, we just do sampling but don’t take expectations.
> - [Assumptions for toy model] For the final linear probing, CL trains a linear layer after feature extractor $g$  for classification. Our toy model analysis illustrates that the CL loss of $g$ is suppressed after SL training with stronger augmentations. It doesn’t matter if there is a linear layer/MLP placed over $g$ in a practical CL model for unsupervised training.  So, it is reasonable to make such assumptions for the toy model in theoretical analysis.
> - [ImageNet-like datasets] We added experiments on ImageNet-100 in Table 3 of the new version. Moreover, Tiny-ImageNet with 64x64 images was also used in [1]. Each class in the train/test set of Mini-ImageNet has 500/100 images, which is the same as Tiny-ImageNet and CIFAR-100.
> | ImageNet-100 | SL   | SimCLR | MoCo | BYOL | SimSiam |
> |--------------|------|--------|------|------|---------|
> | Clean        | 77.8 | 61.8   | 61.8 | 62.2 |    65.8 |
> | AUE          | 5.1  | 5.2    | 6.2  |  7.5 |     4.7 |
> | T-AAP        | 14.4 | 20.3   | 14.5 | 24.8 |    16.6 |
>
> - [Gaps and contrastive unlearnability] Based on suggestions from Reviewer zrxn, we conducted Pearson correlation coefficient analysis of gaps and contrastive unlearnability in Section 4. In Table 1, the PCC between the alignment gap and the SimCLR accuracy is -0.78, and the PCC between the uniformity gap and the SimCLR accuracy is -0.87.  It is revealed that contrastive unlearnability highly relates to alignment and uniformity gaps. Though the statistical metric reflects the relationship, it does not mean accuracy will change monotonically with the gaps.
> On the other hand, in certain settings, the relationship is more clear. In Appendix B.6, we gradually increase the augmentation strength from 0 to the default setting, i.e. s=0.6 in the generation of AUE attacks, and evaluate the alignment and uniformity gaps. In this case, the larger the gaps, the lower the accuracy of SimCLR.
>
> | Strength | Alignment Gap | Uniformity Gap | SimCLR |
> |---------:|---------------:|---------------:|-----------------:|
> |        0 |          0.14 |           0.07 |            83.5 |
> |      0.2 |          0.21 |           0.24 |            64.1 |
> |      0.4 |          0.25 |           0.28 |            56.7 |
> |      0.6 |          0.25 |           0.39 |            52.4 |
>
> - [Alignment] The alignment loss on poisoned data decreases at first while the alignment loss on clean data increases. Thus the alignment gap increases rapidly. Since what we emphasize is the alignment and uniformity gaps which play the role of shortcuts for CL, we replaced Figure 3(c) with a figure of gaps to avoid distraction in the new version. Comparing Figure 3(c) and Figure 4, we find that the trend of CL accuracy aligns with the overall trend of alignment and uniformity gaps. More detailed discussions are added in Section 6.3.
> We look forward to further discussions with you.
>
>
> [1] Rui Wen, Zhengyu Zhao, Zhuoran Liu, Michael Backes, Tianhao Wang, and Yang Zhang. Is adversarial training really a silver bullet for mitigating data poisoning? In International Conference on Learning Representations, 2023

---

> ### Author Response · Authors · 2023-11-21
> **Seeking your feedback**
>
> Dear reviewer,
>
> We hope this message finds you well. We would like to inquire if our responses and explanations in our previous reply have addressed your concerns. Please kindly let us know if there are any further clarifications or modifications that you would like us to make.
> We greatly appreciate your time and effort in providing feedback. Thank you.
>
> Best regards,
>
> The Authors

---

### Official Review · Reviewer_zrxn · 2023-10-30

**Soundness:** 2 fair
**Presentation:** 2 fair
**Contribution:** 2 fair
**Rating:** 3
**Confidence:** 3

**Summary:**

This paper studies unlearnable examples - imperceptible perturbations generated to prevent the released data from unauthorized use. The mechanisms for generating unlearnable examples work similarly to availability attacks. Observing that unlearnable examples generated for supervised learners do not achieve contrastive unlearnability, the paper aims to achieve unlearnability for both supervised and contrastive learning algorithms. Built upon unlearnable example attacks (Huang et al., 2020) and adversarial poisoning attacks (Fowl et al., 2021), the paper proposed to use enhanced data augmentations to create shortcuts for contrastive learning, thus improving the worst-case unlearnability across different supervised and contrastive learning methods.

**Strengths:**

The threat model considers worst-case unlearnability for generating unlearnable examples, which is interesting. In realistic scenarios, the attacker may use any possible learning method to produce a model based on the published unlearnable examples. Therefore, ensuring the data protection scheme of unlearnable examples works for a broad range of learning methods that the attacker may employ is meaningful. The proposed method uses stronger data augmentation, which is straightforward and easy to implement. The paper also provides extensive evaluations regarding the existing methods of availability attacks.

**Weaknesses:**

As shown in the pseudo-code in Section 4.1, the proposed method employs a tuning hyperparameter to control the strength of the data augmentation. While the considered threat model is of practical importance, the technical contributions of the paper are not strong enough.

Another concern is the presented empirical and theoretical results are not structured clearly and coherently, which hinders my understanding of the paper’s overall contributions. For Section 3, the definitions of alignment loss and uniformity loss are introduced in existing work (Wang & Isola, 2020), so they should be moved to the previous background section. The remaining part of Section 3 seems new but is not well-explained. The main empirical finding of Section 3 is that contrastive unlearnability seems correlated with alignment and uniformity gaps. However, there is no clear explanation of how these poisoning methods are grouped in Table 1. It would also be useful to conduct a correlation analysis to demonstrate how strong the correlation is, such as providing the Pearson correlation coefficients. In addition, I do not understand why clean/poisoned alignment & uniformity scores and SL accuracy are also demonstrated in Table 3, which are redundant from my perspective. Moreover, it is hard for me to understand why the results of Table 1 imply the need for enhanced data augmentation. The explanations provided at the end of Section 3 are difficult to parse, and I found the transition between Sections 3 and 4 abrupt. For Section 4, I do not understand the role of Theorem 4.1, where I found the presented theoretical results particularly hard to parse. For example, why do you assume the supervised loss is the mean squared error, and the contrastive loss contains only one negative example? What does the upper bound proven in Theorem 4.1 imply? I would expect a detailed discussion of how Theorem 4.1 connects to the main messages you are trying to convey.

Finally, a minor concern is that the empirical improvements on worst-case unlearnability are not strong. For example, CP-BYOL achieves 41.8% performance on CIFAR-10, which is relatively competitive compared with your methods, while TUE-MoCo achieves relatively similar worst-case unlearnability on CIFAR-100. It would be useful to study why these existing methods can attain good performance and explain how your method improves over them.

**Questions:**

In addition to the questions above, I have the following comments and suggestions for the paper:

1. It would be useful to explain the existing poisoning attacks and their abbreviations in Section 3 before the introduction of Table 1 (instead of Section 5.1). In particular, how these methods are selected and grouped in the table should be explained clearly. Two considered methods, _EntF_ and _HYPO_, are neither effective against _SimCLR_ nor _SL_, so I wonder why they are tested.

2. In the pseudo-code provided in Section 4.1, it is clear that you employ a single parameter _s_ to control the augmentation strength. The parameter applies to three augmentation functions: _RandomsizedCrop()_, _RandomApply()_, and _RandomGrayscale()_. I would like to know whether the worst-case unlearnability can be improved if different hyperparameters are applied to different augmentation functions for your method. A general question is: How does the defender choose the right augmentation functions and their corresponding hyperparameters to achieve the best protection performance?

3. Have you tried to apply your augmentation method to other alternative poisoning attacks, such as TUE and CP? Can you further improve the worst-case learnability based on their method?

4. Section 5.3 presents early stopping as a potential mitigation approach for unlearnable examples. Claiming this as a mitigation method is a bit confusing since unlearnable examples are designed to protect the data from the defender's perspective. It would be helpful to explain in more detail how the attacker can employ early stopping to enhance their attack effectiveness.

5. Tables 3-6 are difficult to read. Please replace them with larger ones in the next version of your paper.

---

> ### Author Response · Authors · 2023-11-16
> **Official Comment by Authors [1/2]**
>
> Thanks for your feedback and questions! We have uploaded a new version of our paper. Specifically, based on your suggestion, we have made some modifications as follows:
> - We investigated the influence of decoupled augmentation factors in Section 6.4.
> - We conducted Pearson correlation coefficient analysis of gaps and contrastive unlearnability in Section 4.  We added more explanation of the CL shortcuts.
> - We relocated the definitions of alignment and uniformity to the ‘Threat Model and Background’ section.
> - We removed the alignment and uniformity losses and preserved only the gaps in Table 1 to avoid redundant information.
> - We moved the tables of  ‘transferability across networks’, ‘poisoning ratio’, and ‘poisoning budget’ to Appendix B.3-B.5.
> - We introduced the attack abbreviations before Table 1.
>
> We summarize other modifications in ‘Author’s comments’.
>
>
> Besides, we’ll make some clarifications as follows:
> 1. [Novelty] We are the first to introduce stronger contrastive augmentations into the poison generation process using supervised models. Our proposed poisoning attacks achieve SOTA worst-case unlearnability across SL and CL algorithms compared to existing methods.
> 2. [Pearson correlation coefficient] In Table 1, the PCC between the alignment gap and the SimCLR accuracy is -0.78, and the PCC between the uniformity gap and the SimCLR accuracy is -0.87.  It is revealed that contrastive unlearnability highly relates to alignment and uniformity gaps.
> 3. [Shortcuts] Table 1 illustrates gaps are the key to shortcuts. It is the reason why AP-based attacks are effective for CL while other non-contrastive poisoning attacks are not. On the other hand, contrastive error-minimization attacks, i.e. CP and TUE naturally possess huge gaps since their generation process optimizes the contrastive loss [1]. It inspires us that if we can mimic the contrastive error-minimization in a supervised way, the resulting attacks possibly obtain both supervised and contrastive unlearnability. In the next section, we analyze how enhanced data augmentations can boost the contrastive unlearnability of basic supervised poisoning approaches.
> 4. [Redundant information in Table 1] We emphasize the alignment and uniformity gaps, so we remove alignment and uniformity losses for a clear presentation. We also include SL accuracy in Table 1 because in later experiments we do not evaluate attacks without CL unlearnability such as UE and DC.
> 5. [Theorem] Recall that contrastive error-minimization replaces the supervised loss in the supervised error-minimization with contrastive loss. The theorem says that the upper bound of contrastive loss decreases with the supervised loss if their data augmentations obey the same distribution. Thus, supervised error minimization with enhanced data augmentations can partially replace the role of contrastive error minimization to enlarge the alignment and uniformity gaps and bring contrastive unlearnability. Moreover, stronger augmentations help adversarial poisoning generate poisons to deceive a contrastive-like reference model and thus improve the contrastive unlearnability as well.
> 6. [Assumption of MSE and InfoNCE] For a toy model analysis, we use MSE for simplicity since CE involves a softmax operation. InfoNCE loss with one negative example is commonly used for theoretical analysis, for example in [2].
> 7. [Comparison with CP and TUE] Our proposed poisoning attacks achieve SOTA worst-case unlearnability across SL and CL algorithms compared to existing methods. Moreover, the unlearnability of CP is not stable: CP-BYOL becomes ineffective for SL on CIFAR-100. On higher-resolution datasets with more classes such as Mini/Tiny-ImageNet, our AUE outperforms TUE-MoCo by 13%/28% which is a great advantage.
> 8. [Augmentations in CP and TUE] These two contrastive error-minimization methods have already involved contrastive training in the poison generation process that employs standard contrastive augmentations (i.e. s=1).  So our methods are not applicable to CP and TUE.
> 9. [HYPO, EntF] These two availability attacks were proposed to deceive adversarial training and degrade the final robust SL accuracy. We test them for a comprehensive evaluation. In the new version, we just group the attacks into contrastive and non-contrastive attacks.
>
>
> ***To be continued.***

---

> > ### Author Response · Authors · 2023-11-16
> > **Official Comment by Authors [2/2]**
> >
> > ***Continuing from the previous comment***
> >
> > 10. [Decoupled augmentations] In Table 4, we decouple the strengths of RandomResizedCrop, RandomColorJitter, and RandomGrayscale and evaluate the CL unlearnability of alternatives. By default, s=0.6 for AUE and s=0.4 for T-AAP. For example, 0-0-s means that RandomResizeCrop strength is 0, RandomColorJitter strength is 0, and RancomGrayscale strength is s. The results illustrate that adjusting these three factors together outperforms adjusting them separately.
> >
> > |       | 0-0-0 | 0-0-s | 0-s-0 | s-0-0 | 0-s-s | s-0-s | s-s-0 | s-s-s |
> > |-------|-------|-------|-------|-------|-------|-------|-------|-------|
> > | AUE   |  83.5 |  58.7 |  79.4 |  88.7 |  60.8 |  56.2 |  87.7 |  52.4 |
> > | T-AAP |  52.3 |  52.0 |  52.9 |  44.9 |  51.4 |  42.2 |  44.8 |  39.1 |
> >
> > 11. [Early stopping] A data publisher protects a dataset using an availability attack. Then, a data collector attempts to mitigate the unlearnability of the poisoned dataset and train a useful model. In Figure 4, at the 5th epoch of SimCLR training, the test accuracy can be higher than the final test accuracy (1000th epoch). Thus, the data collector may choose to stop training at very early epochs. This concept of early stopping has been discussed in [3].
> > We look forward to further discussions with you.
> >
> >
> > [1] Hao He, Kaiwen Zha, and Dina Katabi. Indiscriminate poisoning attacks on unsupervised contrastive learning.
> >
> > [2] Huang W, Yi M, Zhao X, et al. Towards the Generalization of Contrastive Self-Supervised Learning[C]//The Eleventh International Conference on Learning Representations. 2022.
> >
> > [3] Hanxun Huang, Xingjun Ma, Sarah Monazam Erfani, James Bailey, and Yisen Wang. Unlearnable examples: Making personal data unexploitable.

---

> ### Author Response · Authors · 2023-11-21
> **Seeking your feedback**
>
> Dear reviewer,
>
> We hope this message finds you well. We would like to inquire if our responses and explanations in our previous reply have addressed your concerns. Please kindly let us know if there are any further clarifications or modifications that you would like us to make.
> We greatly appreciate your time and effort in providing feedback. Thank you.
>
> Best regards,
>
> The Authors

---

### Official Review · Reviewer_ZSqY · 2023-11-01

**Soundness:** 3 good
**Presentation:** 2 fair
**Contribution:** 3 good
**Rating:** 5
**Confidence:** 5

**Summary:**

Availability attacks aim to safeguard private and commercial datasets from unauthorized use by introducing imperceptible noise and creating unlearnable examples. The goal is to make it extremely challenging for algorithms to train effective models using this data. In cases where supervised learning algorithms fail to achieve this unlearnability, malicious data collectors might turn to contrastive learning algorithms to bypass the protection. Successful attacks must target both supervised and contrastive unlearnability. However, the evaluation shows that most existing availability attacks struggle to achieve contrastive unlearnability, which poses a significant risk to data protection.

This paper reveals that utilizing more robust data augmentations during supervised poisoning generation can lead to the creation of contrastive shortcuts, potentially undermining the protection measures. Leveraging this insight, we introduce AUE and AAP attacks, which significantly enhance worst-case unlearnability across various supervised and contrastive algorithms.

**Strengths:**

1. The performance is commendable and has achieved state-of-the-art results.

2. The paper is well-organized.

**Weaknesses:**

1. There are several typos in the text, such as the need to replace "argmax" with "argmin" in Eq. 1.

2. Consider moving the section on related works from the appendix to the main paper for better visibility and accessibility to readers.

3. Expanding the experiments to include a wider range of methods, such as surrogate-free methods like OPS [1] and robust methods like REM [2], would enhance the comprehensiveness of the evaluation and allow for a more thorough comparison.

4. It would be beneficial to include an evaluation of the attack performance when facing adaptive defenses, such as the inclusion of additional augmentations in the contrastive learning process.

5. Consider conducting experiments on the ImageNet-subset dataset, which includes the first 100 classes of ImageNet data.

6. Consider adding the mean performance value in addition to the worst-case performance in the tables reporting the results.

[1] Shutong Wu, Sizhe Chen, Cihang Xie, and Xiaolin Huang. One-pixel shortcut: On the learning preference of deep neural networks. In Proc. Int’l Conf. Learning Representations, 2023.

[2] Haopeng Fu, Fengxiang He, Yang Liu, Li Shen, and Dacheng Tao. Robust unlearnable examples: Protecting data privacy against adversarial learning. In Proc. Int’l Conf. Learning Representations, 2022

**Questions:**

See weakness above.

---

> ### Author Response · Authors · 2023-11-16
>
> Thanks for your feedback and questions! We have uploaded a new version of our paper. Specifically, based on your suggestion, we have made some modifications as follows:
> - We repositioned the ‘Related Works’ section to the main body of the paper.
> - We rectified some typos.
> - We incorporated the evaluation of OPS and REM attacks in Table 1
> - We included the attack performance on ImageNet-100 in Table 3.
> - We added defenses including ISS-JPEG, ISS-Grayscale, UEraser-Lite, UEraser-Max, and AVATAR in Table 5.
>
> We summarize other modifications in ‘Author’s comments’.
>
>
> Besides, we’ll make some clarifications as follows:
> - [Augmentations in defense] For CL, we have studied additional augmentations in the evaluation stage, including cutout, random noise, and Gaussian blur, which was used in [1] as well. For SL, we included additional experiments of more defenses such as  ISS variants (JPEG, Grayscale), UEraser variants (-Max, -Lite), and AVATAR [2] in Table 5.
> - [Mean performance of unlearnability] Our threat model in Equ (1) considers the worst-case unlearnability since the worst performance across different algorithms determines the reliability of using availability attacks to protect private data, namely the "wooden barrel theory".
> Average unlearnability has two major problems: 1) Fairness. It’s difficult to set proper weights for different evaluation algorithms. For example, it seems to be unfair to take a simple mean of one SL algorithm and multiple CL algorithms. 2) Soundness. Average unlearnability has the potential to underestimate the learnability of the most powerful algorithm.  The reliability of attack methods with the same average unlearnability can vary significantly. For example, under 3 evaluation algorithms, alg A (5%, 10%, 90%)  and alg B (30%, 35%, 40%) have the same average unlearnability of  35%.
> - [ImageNet-100] Our AUE and AAP attacks are also effective on ImageNet-100.
> | ImageNet-100 | SL   | SimCLR | MoCo | BYOL | SimSiam |
> |--------------|------|--------|------|------|---------|
> | Clean        | 77.8 | 61.8   | 61.8 | 62.2 |    65.8 |
> | AUE          | 5.1  | 5.2    | 6.2  |  7.5 |     4.7 |
> | T-AAP        | 14.4 | 20.3   | 14.5 | 24.8 |    16.6 |
>
> We look forward to further discussions with you.
>
> [1] Hao He, Kaiwen Zha, and Dina Katabi. Indiscriminate poisoning attacks on unsupervised contrastive learning.
>
> [2] The Devil's Advocate: Shattering the Illusion of Unexploitable Data using Diffusion Models. https://arxiv.org/abs/2303.08500

---

> > ### Comment · Reviewer_ZSqY · 2023-11-22
> > **Official Comment by Reviewer ZSqY**
> >
> > The response has effectively addressed my concerns. However, I find that augmentation-based enhancement relies more on empirical experiments and lacks a theoretical support. Therefore, I believe this paper still falls around the borderline, and I maintain my rating as 5.

---

> ### Author Response · Authors · 2023-11-21
> **Seeking your feedback**
>
> Dear reviewer,
>
> We hope this message finds you well. We would like to inquire if our responses and explanations in our previous reply have addressed your concerns. Please kindly let us know if there are any further clarifications or modifications that you would like us to make.
> We greatly appreciate your time and effort in providing feedback. Thank you.
>
> Best regards,
>
> The Authors

---

### Official Review · Reviewer_x8t1 · 2023-11-01

**Soundness:** 3 good
**Presentation:** 2 fair
**Contribution:** 2 fair
**Rating:** 5
**Confidence:** 4

**Summary:**

For joint effectiveness of availability attacks on supervised and contrastive learning, the author propose stronger data augmentations to improve worst-case unlearnability on both tasks. Experiments on a range of learning algorithms aim to justify their claim on the proposed method.

**Strengths:**

1. The attacks achieved improved results for multiple supervised and contrastive algorithms.
2. The paper explores the use of label information in poisoning perturbation generation to acquire stable worst-case unlearnability, which contributes to the effectiveness of the proposed attacks.
3. Interesting framing and insights on availability poisoning attacks.

**Weaknesses:**

1. It appears the primary factor contributing to the improvements in results are because of the more aggressive augmentations. It has been known to the community for a while that stronger data augmentations can lead to better defenses against existing unlearnability attacks. ISS and UEraser(-Max) also demonstrated stronger resilience against adaptive attacks in their original papers. It appears that the novelty is diminished slightly by the earlier discoveries regarding stronger augmentations, although they focused on defenses.
2. It remains to be seen whether stronger defense-phase augmentations beats stronger attack-phase augmentations. The few results on this is in Table 3, and the answer remains inconclusive. The reviewer suspects that “shortcuts” are difficult to form with such stronger defenses with a tight perturbation budget.
3. In Figure 1, UT-AAP is not strictly better than UT-AP.
4. It is confusing why Sections 4.2 and 4.3 are separate. The difference between AUE, AAP exists only in the use of existing error-minimizing and maximizing objectives employed in existing attacks, which are not the core contribution of the paper. It would be better to refactor Section 4 to combine both algorithms and sections for clarity, as it appears redundant in the current format.

**Questions:**

Potential Improvements:
1. Please consider adding more defense baselines against the proposed attacks, e.g. the ISS variants, UEraser variants, and AVATAR [1].
2. The motivation for improved / stronger augmentation (the key contribution of this paper) should be further strengthened. The rationale behind Theorem 4.1 could benefit from additional clarification, as its purpose remains somewhat ambiguous. In Section 4.1, the primary takeaway appears to be the notion that to effectively generate poisons, it is imperative to employ stronger augmentations in line with those utilized by contrastive learning algorithms.
3. It would be better to consider ImageNet-100 instead of Mini-ImageNet to align with previous work.
4. A discussion is needed on the proposed method, TUE and CP (the most relevant baselines), especially from the perspective of computation overheads, transferability, etc.

Minor Issues:
1. Avoid breaking Code Listing 1 between two pages.
2. Tables 3-6 are way too small.
3. "obey the same distribution. ." -> "obey the same distribution."

[1] The Devil's Advocate: Shattering the Illusion of Unexploitable Data using Diffusion Models. https://arxiv.org/abs/2303.08500

---

> ### Author Response · Authors · 2023-11-16
>
> Thanks for your feedback and questions!
> We have uploaded a new version of our paper. In particular, considering your valuable suggestions, we have implemented the following modifications:
> - We added defenses including ISS-JPEG, ISS-Grayscale, UEraser-Lite, UEraser-Max, and AVATAR in Table 5.
> - We included the attack performance on ImageNet-100 in Table 3.
> - We moved the pseudo-code of augmentations to Appendix A.2.
> - We moved the tables of  ‘transferability across networks’, ‘poisoning ratio’, and ‘poisoning budget’ to Appendix B.3-B.5.
> - We included the comparison of computation consumption with CP and TUE in Appendix B.1.
>
> We summarize other modifications in ‘Author’s comments’.
>
> Besides, we’ll make some clarifications as follows:
> - [Novelty] We are the first to introduce stronger contrastive augmentations into the poison generation process using supervised models. Our proposed poisoning attacks achieve SOTA worst-case unlearnability across SL and CL algorithms compared to existing methods.
> - [Stronger defenses] According to Table 5 in the new version, UEraser-Max demonstrates stronger strength compared to other UEraser variants, resulting in improved defense performance. However, UEraser-Max is not particularly effective against AUE attacks. Grayscale proves ineffective, while JPEG surpasses ISS in performance. AVATAR achieves the highest defense performance for our attacks, reaching approximately 85-88%.
> - [UT-AAP and UT-AP] The utilization of stronger augmentations in our methods enhances contrastive unlearnability, but it may potentially weaken supervised unlearnability. This compromise is further complicated by the inherent instability in generating untargeted adversarial poisoning [1].
> - [AUE and AAP] The roles of augmentations in AUE and AAP are somewhat different. In the case of AUE, supervised error minimization with enhanced data augmentations can partially replace the role of contrastive error minimization to enlarge the alignment and uniformity gaps and bring contrastive unlearnability. Moreover, stronger augmentations help adversarial poisoning generate poisons to deceive a contrastive-like reference model and thus improve the contrastive unlearnability as well.
> - [Theorem] Recall that contrastive error-minimization replaces the supervised loss in the supervised error-minimization with contrastive loss. The theorem says that the upper bound of contrastive loss decreases with the supervised loss if their data augmentations in SL obey the same distribution as CL. Thus, optimizing the augmented supervised loss in the poison generation process implicitly optimizes the contrastive loss.
> - [ImageNet-100] Our AUE and AAP attacks are also effective on ImageNet-100.
> | ImageNet-100 | SL   | SimCLR | MoCo | BYOL | SimSiam |
> |--------------|------|--------|------|------|---------|
> | Clean        | 77.8 | 61.8   | 61.8 | 62.2 |    65.8 |
> | AUE          | 5.1  | 5.2    | 6.2  |  7.5 |     4.7 |
> | T-AAP        | 14.4 | 20.3   | 14.5 | 24.8 |    16.6 |
> - [Computation overheads] Computation efficiency is important for protecting private data using availability attacks in real-world scenarios. On CIFAR-10/100 and using a single NVIDIA 3090 GPU, AUE/AAP costs around 2.7/2.2 hours while CP-SimCLR costs around 48 hours, and TUE-MoCo costs around 8.5 hours to generate poisons. Our supervised training-based attacks are much more efficient than contrastive training-based ones. The results of our methods on other datasets are shown in Appendix B.1.
>
> We look forward to further discussions with you.
>
>
> [1] Fowl, Liam, et al. "Adversarial examples make strong poisons." Advances in Neural Information Processing Systems 34 (2021): 30339-30351.

---

> ### Author Response · Authors · 2023-11-21
> **Seek your feedback**
>
> Dear reviewer,
>
> We hope this message finds you well. We would like to inquire if our responses and explanations in our previous reply have addressed your concerns. Please kindly let us know if there are any further clarifications or modifications that you would like us to make.
> We greatly appreciate your time and effort in providing feedback. Thank you.
>
> Best regards,
>
> The Authors

---

### Official Review · Reviewer_6pFa · 2023-11-03

**Soundness:** 4 excellent
**Presentation:** 3 good
**Contribution:** 3 good
**Rating:** 6
**Confidence:** 4

**Summary:**

The authors explore the problem of unlearnable data in an unsupervised setting (in addition to the more common supervised setting). The authors employ stronger data augmentations in their proposed attacks to boost the potency of the unlearnable samples in the unsupervised domain.

**Strengths:**

* The authors do a very thorough job in their experimentation and literature review.
* The problem is well motivated, and the authors explore it in a principled and thorough way.
* The work seems to demonstrate a Pareto improvement over existing methods meant to generate poisons for unsupervised and supervised learning.

**Weaknesses:**

*  An existing method SEP seems to demonstrate a Pareto improvement over every attack this work proposes, except for UT-AAP.
* I do appreciate the authors principled approach, and intuition, and analysis. Although I think the finding that strong augmentations during poison crafting improves unlearnable examples was also found in [1]. But this work does include very thorough analysis of this, as well as introducing more augmentations during training of the generating model.
* This isn't a weakness of the core work, but tables 3-6 are way too small and it can be aggravating to have to zoom in significantly while reading the work. I would suggest moving to the appendix. Also legends/labels on several of the figures had this same problem.


[1] Fowl, Liam, et al. "Adversarial examples make strong poisons." Advances in Neural Information Processing Systems 34 (2021): 30339-30351.

**Questions:**

* Should the constraints in Eq 1 be $f_\delta \in \text{argmin}_f ...$ instead?

---

> ### Author Response · Authors · 2023-11-16
>
> Thank you for your recognition and feedback!  We have uploaded a new version of our paper. Specifically, based on your suggestions, we have moved some tables to the appendix and rectified the typos. We summarize other modifications in ‘Author’s comments’.
>
>
> Besides, we’ll make some clarifications as follows:
> - [About SEP] On CIFAR-10, SEP is not a Pareto improvement of AUE since for SimSiam evaluation, AUE outperforms SEP (34.5% v.s 36.7%).
> Furthermore, on CIFAR-100, both AUE and T-AAP attacks outperform SEP in the worst-case unlearnability.
> - [Augmentations] The motivation, technique, and goal of augmentations in our methods are different from [1]. Differential augmentations employed by [1] contain only standard SL augmentations including RandomCrop and RandomHorizontalFlip. However, our method is motivated by empirical observations and theoretical analysis that combining supervised poisoning approaches with much stronger contrastive augmentations can craft effective poisons for both SL and CL.
>
> We look forward to further discussions with you.

---

> ### Author Response · Authors · 2023-11-21
> **Seeking your feedback**
>
> Dear reviewer,
>
> We hope this message finds you well. We would like to inquire if our responses and explanations in our previous reply have addressed your concerns. Please kindly let us know if there are any further clarifications or modifications that you would like us to make.
> We greatly appreciate your time and effort in providing feedback. Thank you.
>
> Best regards,
>
> The Authors

---

### Author Response · Authors · 2023-11-16
**Summary of changes in new version**

Thanks for the reviewers' constructive comments. Based on their feedback and suggestions, we have uploaded a new version which includes the following changes compared to the initial version:

Changes in Structure:
- We repositioned the ‘Related Works’ section to the main body of the paper.
- We relocated the definitions of alignment and uniformity to the ‘Threat Model and Background’ section.
- We moved the tables of  ‘transferability across networks’, ‘poisoning ratio’, and ‘poisoning budget’ to Appendix B.3-B.5.
- We moved the pseudo-code of augmentations to Appendix A.2.
- We introduced the attack abbreviations before Table 1.
- We included the comparison of computation consumption with CP and TUE in Appendix B.1.

Additions to the experiments:
- We included the attack performance on ImageNet-100 in Table 3.
- We incorporated the evaluation of OPS and REM attacks in Table 1.
- We added defenses including ISS-JPEG, ISS-Grayscale, UEraser-Lite, UEraser-Max, and AVATAR in Table 5.
- We investigated the influence of decoupled augmentation factors in Section 6.4.
- We evaluated the alignment and uniformity gaps and SimCLR accuracy with different strengths for AUE in Appendix B.6.

Presentation improvement:
- We conducted Pearson correlation coefficient analysis of gaps and contrastive unlearnability in Section 4.  We added more explanation of the CL shortcuts.
- We removed the alignment and uniformity losses and preserved only the gaps in Table 1 to avoid redundant information.
- We displayed the gaps instead of losses during training in Figure 3(c). Additionally, we relocated the figure illustrating the impact of strength in T-AAP to Figure 3(d).
- We enhanced the formatting of two algorithms.
- We rectified some typos.

---

### Author Response · Authors · 2023-11-21
**A reminder**

Dear reviewers,

As the discussion phase is coming to a close, we would like to draw your attention to the changes we have made in the new version. Based on your feedback, we have included additional experiments and provided further clarification and explanations.
We look forward to discussing with you further.
Thank you.



Best regards,

The Authors

---

### Meta-Review · Area_Chair_JEbW · 2023-12-06

**Metareview:**

The paper proposes a new algorithm for generating unlearnable examples with augmentations against supervised learning and contrastive learning.

Strengths:

1.  The author tries to label information in poisoning perturbation generation to improve unlearnability, and the results show their method's effectiveness.

Weaknesses:
1. The paper still has problems with hyperparameter choosing.
2. The analysis is not consistent with their practical methods.

Therefore, I agree with most of the reviewers and reject this paper.

**Justification For Why Not Higher Score:**

N/A

**Justification For Why Not Lower Score:**

N/A

---

### Decision · Program_Chairs · 2024-01-16

Reject